# Towards Generative Abstract Reasoning: Completing Raven's Progressive Matrix via Rule Abstraction and Selection

**Fan Shi**    **Bin Li**[*]   **Xiangyang Xue**
Shanghai Key Laboratory of Intelligent Information Processing
School of Computer Science, Fudan University
`fshi22@m.fudan.edu.cn`   `{libin,xyxue}@fudan.edu.cn`

## Abstract

Endowing machines with abstract reasoning ability has been a long-term research topic in artificial intelligence. Raven's Progressive Matrix (RPM) is widely used to probe abstract visual reasoning in machine intelligence, where models will analyze the underlying rules and select one image from candidates to complete the image matrix. Participators of RPM tests can show powerful reasoning ability by inferring and combining attribute-changing rules and imagining the missing images at arbitrary positions of a matrix. However, existing solvers can hardly manifest such an ability in realistic RPM tests. In this paper, we propose a deep latent variable model for answer generation problems through **R**ule **A**bstract**I**on and **SE**lection (RAISE). RAISE can encode image attributes into latent concepts and abstract atomic rules that act on the latent concepts. When generating answers, RAISE selects one atomic rule out of the global knowledge set for each latent concept to constitute the underlying rule of an RPM. In the experiments of bottom-right and arbitrary-position answer generation, RAISE outperforms the compared solvers in most configurations of realistic RPM datasets. In the odd-one-out task and two held-out configurations, RAISE can leverage acquired latent concepts and atomic rules to find the rule-breaking image in a matrix and handle problems with unseen combinations of rules and attributes.

## 1 Introduction

The abstract reasoning ability is pivotal to abstracting the underlying rules from observations and quickly adapting to novel situations (Cattell, 1963; Zhuo & Kankanhalli, 2021; Małkiński & Mańdziuk, 2022a), which is the foundation of cognitive processes (Gray & Thompson, 2004) like number sense (Dehaene, 2011), spatial reasoning (Byrne & Johnson-Laird, 1989), and physical reasoning (McCloskey, 1983). Intelligent systems may benefit from human-like abstract reasoning when leveraging acquired skills in unseen tasks (Barrett et al., 2018), for example, generalizing the law of object collision in the simulation environment to real scenes. Therefore, endowing intelligent systems with abstract reasoning ability is the cornerstone of higher-intelligence systems and a long-lasting research topic of artificial intelligence (Chollet, 2019; Małkiński & Mańdziuk, 2022b).

Raven's Progressive Matrix (RPM) is a classical test of abstract reasoning ability for human and intelligent systems (Małkiński & Mańdziuk, 2022a), where participators need to choose one image out of eight candidates to fill in the bottom-right position of a 3×3 image matrix (Raven & Court, 1998). Previous studies demonstrate that participators can display powerful reasoning ability by directly imagining the missing images (Hua & Kunda, 2020; Pekar et al., 2020), and answer-generation tasks can more accurately reflect the model's understanding of underlying rules than answer-selection ones (Mitchell, 2021). For example, some RPM solvers find shortcuts in discriminative tasks by selecting answers according to the bias of candidate sets instead of the given context.

To solve answer-selection problems, many solvers fill each candidate to the matrix for score estimation and can hardly imagine answers from the given context (Barrett et al., 2018; Hu et al., 2021).

---

[*]Corresponding author

Some generative solvers have been proposed to solve answer-generation tasks (Pekar et al., 2020; Zhang et al., 2021b;a). They generate solutions for bottom-right images and select answers by comparing the solutions and candidates. However, some generative solvers do not parse interpretable attributes and attribute-changing rules from RPMs (Pekar et al., 2020), and usually introduce artificial priors in the processes of representation learning or abstract reasoning (Zhang et al., 2021b;a). On the other hand, most generative solvers are trained with the aid of candidate sets in training, bringing the potential risk of learning shortcuts (Hu et al., 2021; Benny et al., 2021).

Deep latent variable models (DLVMs) (Kingma & Welling, 2013; Sohn et al., 2015) can capture underlying structures of noisy observations via interpretable latent spaces (Edwards & Storkey, 2017; Eslami et al., 2018; Garnelo et al., 2018; Kim et al., 2019). Previous work (Shi et al., 2021) solves generative RPM problems by regarding attributes and attribute-changing rules as latent concepts, which can generate solutions by executing attribute-specific predictive processes. Through conditional answer-generation processes that consider the underlying structure of RPM panels, the distractors are not necessary to train DLVM-based solvers. Although previous work has achieved answer generation in RPMs with continuous attributes, understanding complex discrete rules and abstracting global rules in realistic datasets is still challenging for DLVMs.

This paper proposes a DLVM for generative RPM problems through **R**ule **A**bstract**I**on and **SE**lection (RAISE) [1]. RAISE encodes image attributes (e.g., object size and shape) as independent latent concepts to bridge high-dimensional images and latent representations of rules. The underlying rules of RPMs are decomposed into subrules in terms of latent concepts and abstracted into atomic rules as a set of learnable parameters shared among RPMs. RAISE picks up proper rules for each latent concept and combines them into the integrated rule of an RPM to generate the answer. The conditional generative process of RAISE indicates how to use the global knowledge of atomic rules to imagine (generate) target images (answers) *interpretably*. RAISE can automatically parse latent concepts without meta information of image attributes to reduce artificial priors in the learning process. RAISE can be trained under semi-supervised settings, requiring only a small amount of rule annotations to outperform the compared models in non-grid configurations. By predicting the target images at arbitrary positions, RAISE does not require distractors of candidate sets in training and supports generating missing images at arbitrary and even multiple positions.

RAISE outperforms the compared solvers when generating bottom-right and arbitrary-position answers in most configurations of datasets. We interpolate and visualize the learned latent concepts and apply RAISE in odd-one-out problems to demonstrate its interpretability. The experimental results show that RAISE can detect the rule-breaking image of a matrix through interpretable latent concepts. Finally, we evaluate RAISE on two out-of-distribution configurations where RAISE retains relatively higher accuracy when encountering unseen combinations of rules and attributes.

## 2 RELATED WORK

**Generative RPM Solvers.** While selective RPM solvers (Zhuo & Kankanhalli, 2021; Barrett et al., 2018; Wu et al., 2020; Hu et al., 2021; Benny et al., 2021; Steenbrugge et al., 2018; Hahne et al., 2019; Zhang et al., 2019b; Zheng et al., 2019; Wang et al., 2019; 2020; Jahrens & Martinetz, 2020) focus on answer-selection problems, generative solvers predict representations or images at missing positions (Pekar et al., 2020; Zhang et al., 2021b;a). Niv et al. extract image representations through Variational AutoEncoder (VAE) (Kingma & Welling, 2013) and design a relation-wise perception process for answer prediction (Pekar et al., 2020). With interpretable scene representations, ALANS (Zhang et al., 2021b) and PrAE (Zhang et al., 2021a) adopt algebraic abstract and symbolic logical systems as the reasoning backends. These generative solvers predict answers at the bottom-right position. LGPP (Shi et al., 2021) and CLAP (Shi et al., 2023) learn hierarchical latent variables to capture the underlying rules of RPMs with random functions (Williams & Rasmussen, 2006; Garnelo et al., 2018), and can generate answers at arbitrary positions on RPMs with continuous attributes. RAISE is a variant of DLVM to realize generative abstract reasoning on realistic RPM datasets with discrete attributes and rules through atomic rule abstraction and selection.

**Bayesian Inference with Global Latent Variables.** DLVMs (Kingma & Welling, 2013; Sohn et al., 2015; Sønderby et al., 2016) can capture underlying structures of high-dimensional data in latent

---

[1]Code is available at https://github.com/FudanVI/generative-abstract-reasoning/tree/main/raise

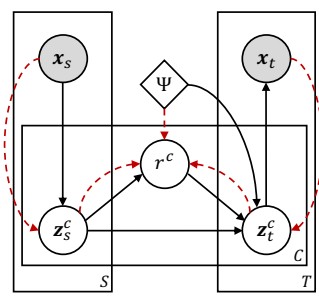

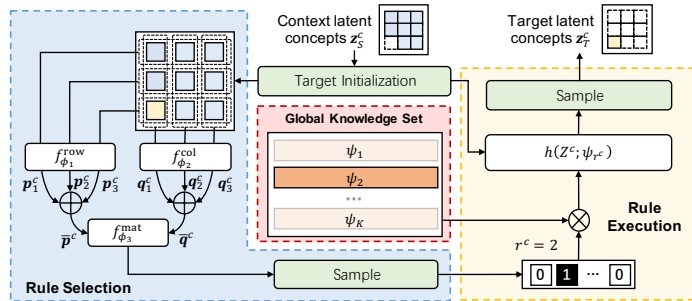

(a) Graphical model of RAISE  (b) Abstract reasoning process

Figure 1: **An overview of RAISE**. The graphical model in (a) displays the generative process (solid black lines) and inference process (dashed red lines). Panel (b) shows the computational details of the abstract reasoning process and highlights the rule selection, rule execution, and global knowledge with blue, yellow, and red backgrounds, respectively.

spaces, regard shared concepts as global latent variables, and introduce local latent variables conditioned on the shared concepts to distinguish each sample. GQN (Eslami et al., 2018) captures entire 3D scenes via global latent variables to generate 2D images of unseen perspectives. With object-centric representations (Yuan et al., 2023), global latent variables can explain layouts of scenes (Jiang & Ahn, 2020) or object appearances for multiview scene generation (Chen et al., 2021; Kabra et al., 2021; Yuan et al., 2022; Gao & Li, 2023; Yuan et al., 2024). Global concepts can describe common features of elements in data with exchange invariance like sets (Edwards & Storkey, 2017; Hewitt et al., 2018; Giannone & Winther, 2021). NP family (Garnelo et al., 2018; Kim et al., 2019; Foong et al., 2020) constructs different function spaces through global latent variables. DLVMs can generate answers at arbitrary positions of an RPM by regarding the concept-changing rules as global concepts (Shi et al., 2021; 2023). RAISE holds a similar idea of modeling underlying rules as global concepts. Unlike previous works, RAISE attempts to abstract the atomic rules shared among RPMs.

## 3 METHOD

In this paper, an RPM problem is $(\boldsymbol{x}_S, \boldsymbol{x}_T)$ where $\boldsymbol{x}_S$ and $\boldsymbol{x}_T$ are mutually exclusive sets of images, $S$ indexes the given context images, and $T$ indexes the target images to predict ($T$ can index multiple images). The objective of RAISE is to maximize the log-likelihood $\log p(\boldsymbol{x}_T | \boldsymbol{x}_S)$ while learning atomic rules shared among RPMs. In the following sections, we will introduce the generative and inference processes of RAISE that can abstract and select atomic rules in the latent space.

### 3.1 CONDITIONAL GENERATION

The generative process is the foundation of answer generation, including the stages of concept learning, abstract reasoning, and image generation.

**Concept Learning.** RAISE extracts interpretable image representations for abstract reasoning and image generation in the concept learning stage. Previous studies have emphasized the role of abstract object representations in the abstract reasoning of infants (Kahneman et al., 1992; Gordon & Irwin, 1996) and the benefit of disentangled representations for RPM solvers (Van Steenkiste et al., 2019), which reflect the compositionality of human cognition (Lake et al., 2011). RAISE realizes compositionality by learning latent representations of attributes (Shi et al., 2021; 2023). RAISE regards image attributes as latent concepts and decomposes the rules of RPMs into atomic rules based on the latent concepts. Since the description of attributes is not provided in training, the latent concepts learned by RAISE are not exactly the same as the realistic attributes defined in the dataset. RAISE extracts $C$ context latent concepts $\boldsymbol{z}_s = \{\boldsymbol{z}_s^c\}_{c=1}^C$ for each context image $\boldsymbol{x}_s$ ($s \in S$):

$$\boldsymbol{\mu}_s^{1:C} = g_\theta^{\text{enc}}(\boldsymbol{x}_s), \qquad\qquad s \in S,$$
$$\boldsymbol{z}_s^c \sim \mathcal{N}(\boldsymbol{\mu}_s^c, \sigma_z^2 \boldsymbol{I}), \qquad c = 1, .., C, \quad s \in S. \tag{1}$$

The encoder $g_\theta^{\text{enc}}$ outputs the mean of context latent concepts. The standard deviation is controlled by a hyperparameter $\sigma_z$ to keep training stability. Each context image is processed through $g_\theta^{\text{enc}}$

independently, making it possible to extract latent concepts for any set of input images. In this stage, the encoder does not consider any relationships between images and focuses on concept learning.

**Abstract Reasoning.** As illustrated in Figure 1b, RAISE predicts target latent concepts $\boldsymbol{z}_T$ from context latent concepts $\boldsymbol{z}_S$ in the abstract reasoning stage, involving rule abstraction, rule selection, and rule execution processes. To abstract atomic rules and build the global knowledge set, RAISE adopts $K$ global learnable parameters $\psi = \{\psi_k\}_{k=1}^K$, each indicating an atomic rule shared among RPMs. In rule selection, we use categorical indicators $\{r^c\}_{c=1}^C$ ($r^c = 1, ..., K$) to select a proper rule out of $\psi$ for each concept. Inferring the indicators from $\boldsymbol{z}_S$ correctly is critical to rule selection. RAISE creates a $3\times3$ representation matrix $\boldsymbol{Z}^c$ for each concept, initializing the representations of context images with the corresponding context latent concepts and those of target images with zero vectors. Then RAISE extracts the row-wise and column-wise representations:

$$\boldsymbol{p}_i^c = f_{\phi_1}^{\mathrm{row}}\left(\boldsymbol{Z}_{i,1:3}^c\right), \quad \boldsymbol{q}_i^c = f_{\phi_2}^{\mathrm{col}}\left(\boldsymbol{Z}_{1:3,i}^c\right), \quad i = 1, 2, 3, \quad c = 1, ..., C. \tag{2}$$

RAISE averages the representations via $\bar{\boldsymbol{p}}^c = (\boldsymbol{p}_1^c + \boldsymbol{p}_2^c + \boldsymbol{p}_3^c)/3$ and $\bar{\boldsymbol{q}}^c = (\boldsymbol{q}_1^c + \boldsymbol{q}_2^c + \boldsymbol{q}_3^c)/3$ to obtain integrated representations of row and column rules. We concatenate $\bar{\boldsymbol{p}}^c$ and $\bar{\boldsymbol{q}}^c$ to acquire the probability of selecting atomic rules out of the global knowledge set:

$$r^c \sim \mathrm{Categorical}\left(\boldsymbol{\pi}_{1:K}^c\right), \quad \pi_1^c, ..., \pi_K^c = f_{\phi_3}^{\mathrm{ind}}(\bar{\boldsymbol{p}}^c, \bar{\boldsymbol{q}}^c), \quad c = 1, ..., C. \tag{3}$$

We denote the learnable parameters as $\phi = \{\phi_1, \phi_2, \phi_3\}$ for convenience. In rule execution, RAISE selects and executes an atomic rule on each concept to predict the target latent concepts:

$$\begin{aligned} \boldsymbol{\mu}_T^c &= h\left(\boldsymbol{Z}^c; \psi_{r^c}\right), & c = 1, ..., C, \\ \boldsymbol{z}_t^c &\sim \mathcal{N}\left(\boldsymbol{\mu}_t^c, \sigma_z^2 \boldsymbol{I}\right), & t \in T, \quad c = 1, ..., C. \end{aligned} \tag{4}$$

RAISE instantiates $h$ by selecting the $r^c$-th learnable parameters from the global knowledge set $\psi$ to convert the zero-initialized target representations in $\boldsymbol{Z}^c$ into the mean of target latent concepts. As in the concept learning stage, the standard deviation of target latent concepts is controlled by $\sigma_z$. $h$ consists of convolution layers to aggregate information from neighbor context latent concepts on the matrix and update target latent concepts. Each learnable parameters in $\psi$ indicates a type of atomic rule. See Appendix C.1 for the detailed description of $h$.

**Image Generation.** Finally, RAISE decodes the target latent concepts predicted in the abstract reasoning stage into the mean of target images:

$$\boldsymbol{x}_t \sim \mathcal{N}\left(\boldsymbol{\Lambda}_t, \sigma_x^2 \boldsymbol{I}\right), \quad \boldsymbol{\Lambda}_t = g_\varphi^{\mathrm{dec}}\left(\boldsymbol{z}_t^{1:C}\right), \quad t \in T. \tag{5}$$

RAISE generates each target image independently to make the decoder focus on image reconstruction. We control the noise of target images by setting the standard deviation $\sigma_x$ as a hyperparameter.

According to Figure 1a, we decompose the conditional generative process as

$$p_\Theta(\boldsymbol{h}, \boldsymbol{x}_T | \boldsymbol{x}_S) = \prod_{t \in T} p_\varphi(\boldsymbol{x}_t | \boldsymbol{z}_t) \prod_{c=1}^C \left( p_\psi(\boldsymbol{z}_T^c | r^c, \boldsymbol{z}_S^c) p_\phi(r^c | \boldsymbol{z}_S^c) \prod_{s \in S} p_\theta(\boldsymbol{z}_s^c | \boldsymbol{x}_s) \right) \tag{6}$$

where $\boldsymbol{h}$ is the set of all latent variables and $\Theta = \{\theta, \phi, \psi, \varphi\}$ are learnable parameters of RAISE.

## 3.2 Variational Inference

RAISE approximates the untractable posterior with a variational distribution $q(\boldsymbol{h} | \boldsymbol{x}_T, \boldsymbol{x}_S)$ (Kingma & Welling, 2013), which consists of the following distributions.

$$\begin{aligned} q(\boldsymbol{z}_s^c | \boldsymbol{x}_s) &= \mathcal{N}\left(\tilde{\boldsymbol{\mu}}_s^c, \sigma_z^2 \boldsymbol{I}\right), & s \in S, \quad c = 1, ..., C, \\ q(\boldsymbol{z}_t^c | \boldsymbol{x}_t) &= \mathcal{N}\left(\tilde{\boldsymbol{\mu}}_t^c, \sigma_z^2 \boldsymbol{I}\right), & t \in T, \quad c = 1, ..., C, \\ q(r^c | \boldsymbol{z}_S^c, \boldsymbol{z}_T^c) &= \mathrm{Categorical}\left(\tilde{\boldsymbol{\pi}}_{1:K}^c\right), & c = 1, ..., C. \end{aligned} \tag{7}$$

Since RAISE shares the encoder between the generative and inference processes to reduce the model parameters, we compute context latent concepts $\tilde{\boldsymbol{\mu}}_s^{1:C}$ and target latent concepts $\tilde{\boldsymbol{\mu}}_t^{1:C}$ via the same process described in Equation 1. In the inference process, RAISE reformulates the variational distribution of the categorical indicator $r^c$ as $q(r^c | \boldsymbol{z}_S^c, \boldsymbol{z}_T^c) \propto p(\boldsymbol{z}_T^c | r^c, \boldsymbol{z}_S^c) p(r^c | \boldsymbol{z}_S^c)$. That is, RAISE

predicts the prior probabilities $\boldsymbol{\pi}_{1:K}^c$ of $p(r^c|\boldsymbol{z}_S^c)$ from the context latent concepts $\boldsymbol{z}_S^c$ and compute the likelihood $p(\boldsymbol{z}_T^c|r^c, \boldsymbol{z}_S^c)$ by executing the atomic rule $r^c$ ($r^c = 1, \cdots, K$) on $\boldsymbol{z}_S^c$. In this way, we can estimate the variational distribution $q(r^c|\boldsymbol{z}_S^c, \boldsymbol{z}_T^c)$ by considering both the prior probabilities and the likelihoods of $K$ atomic rules, which reduces the risk of model collapse (e.g., always selecting one atomic rule from $\psi$). We provide more details of $q(r^c|\boldsymbol{z}_S^c, \boldsymbol{z}_T^c)$ in Appendix A.1. Letting $\Psi = \{\theta, \phi, \psi\}$, we factorize the variational distribution as

$$q_\Psi(\boldsymbol{h}|\boldsymbol{x}_T, \boldsymbol{x}_S) = \prod_{c=1}^{C} \left( q_{\phi,\psi}(r^c|\boldsymbol{z}_S^c, \boldsymbol{z}_T^c) \prod_{s \in S} q_\theta(\boldsymbol{z}_s^c|\boldsymbol{x}_s) \prod_{t \in T} q_\theta(\boldsymbol{z}_t^c|\boldsymbol{x}_t) \right). \tag{8}$$

## 3.3 Parameter Learning

We update the parameters of RAISE by maximizing the evidence lower bound (ELBO) of the log-likelihood $\log p(\boldsymbol{x}_T|\boldsymbol{x}_S)$ (Kingma & Welling, 2013). With the generative process $p_\Theta$ and the variational distribution $q_\Psi$ defined in Equations 6 and 8, the ELBO is ($q$ denotes the variational distribution, and we omit the parameter symbols $\Theta$ and $\Psi$ for convenience)

$$\mathcal{L} = \mathbb{E}_{q_\Psi(\boldsymbol{h}|\boldsymbol{x}_T, \boldsymbol{x}_S)} \left[ \log \frac{p_\Theta(\boldsymbol{h}, \boldsymbol{x}_T|\boldsymbol{x}_S)}{q_\Psi(\boldsymbol{h}|\boldsymbol{x}_T, \boldsymbol{x}_S)} \right]$$

$$= \underbrace{\sum_{t \in T} \mathbb{E}_q \left[ \log p(\boldsymbol{x}_t|\boldsymbol{z}_t) \right]}_{\mathcal{L}_{\text{rec}}} - \underbrace{\sum_{c=1}^{C} \mathbb{E}_q \left[ \log \frac{q(\boldsymbol{z}_T^c|\boldsymbol{x}_T)}{p(\boldsymbol{z}_T^c|r^c, \boldsymbol{z}_S^c)} \right]}_{\mathcal{R}_{\text{pred}}} - \underbrace{\sum_{c=1}^{C} \mathbb{E}_q \left[ \log \frac{q(r^c|\boldsymbol{z}_S^c, \boldsymbol{z}_T^c)}{p(r^c|\boldsymbol{z}_S^c)} \right]}_{\mathcal{R}_{\text{rule}}} \tag{9}$$

The **reconstruction loss** $\mathcal{L}_{\text{rec}}$ measures the quality of the reconstruction images. The **concept regularizer** $\mathcal{R}_{\text{pred}}$ estimates the distance between the predicted target concepts and the concepts directly encoded from target images. Minimizing $\mathcal{R}_{\text{pred}}$ will promote RAISE to generate correct predictions in the space of latent concepts. The **rule regularizer** $\mathcal{R}_{\text{rule}}$ expects RAISE to select the same rules when given different sets of images in an RPM. The variational posterior $q(r^c|\boldsymbol{z}_S^c, \boldsymbol{z}_T^c)$ conditioned on the entire matrix and the prior $p(r^c|\boldsymbol{z}_S^c)$ conditioned on the context images are expected to have similar probabilities. The detailed derivation of the ELBO is provided in Appendix A.2.

The abstraction and selection of atomic rules rely on the acquired latent concepts. Therefore, RAISE introduces auxiliary rule annotations to improve the quality of latent concepts and stabilize the learning process. We denote rule annotations as $\boldsymbol{v} = \{v_a\}_{a=1}^A$ where $A$ is the number of ground truth attributes and $v_a$ indicates the type of rules on the $a$-th attribute. For example, $\boldsymbol{v} = [2, 1, 3]$ means that the attributes follow the second, first, and third rules respectively. RAISE does not leverage the meta-information of attributes in training since the rule annotations only inform the type of rule on each attribute. The meaning of attributes is automatically learned by RAISE for accurate rule abstraction and selection. One key to guiding concept learning with rule annotations is determining the correspondence between latent concepts and attributes. RAISE introduces a $A \times C$ binary matrix $\boldsymbol{M}$ where $\boldsymbol{M}_{a,c} = 1$ indicates that the $a$-th attribute is encoded in the $c$-th latent concept. Therefore, the rule predicted on the $c$-th latent concept is supervised by the rule annotation $v_a$, and the auxiliary loss measures distances between the predicted and ground truth types of rules:

$$\mathcal{L}_{\text{sup}} = \frac{1}{2} \sum_{a=1}^{A} \sum_{c=1}^{C} \boldsymbol{M}_{a,c} \log \left( \pi_{v_a}^c + \tilde{\pi}_{v_a}^c \right). \tag{10}$$

The auxiliary loss $\mathcal{L}_{\text{sup}}$ is the log-likelihood of the categorical distributions considering the attribute-concept correspondence $\boldsymbol{M}$. The binary matrix $\boldsymbol{M}$ is derived by solving the following assignment problem on a batch of RPM samples:

$$\underset{\boldsymbol{M}}{\arg\max} \ \mathcal{L}_{\text{sup}} \quad \text{s.t.} \quad \begin{cases} \sum_{c=1}^{C} \boldsymbol{M}_{a,c} = 1, & a = 1, ..., A, \\ \sum_{a=1}^{A} \boldsymbol{M}_{a,c} = 0 \text{ or } 1, & c = 1, ..., C, \\ \boldsymbol{M}_{a,c} = 0 \text{ or } 1, & a = 1, ..., A, \quad c = 1, ..., C. \end{cases} \tag{11}$$

Equation 11 allows the existence of redundant latent concepts, which can be solved using the modified Jonker-Volgenant algorithm (Crouse, 2016). In this case, the training objective becomes

$$\underset{\Theta}{\arg\max} \ \mathcal{L}_{\text{rec}} - \beta_1 \mathcal{R}_{\text{pred}} - \beta_2 \mathcal{R}_{\text{rule}} + \beta_3 \mathcal{L}_{\text{sup}} \tag{12}$$

Table 1: **The accuracy (%) of selecting bottom-right answers on different configurations (i.e., *Center*, *L-R*, etc) of RAVEN/I-RAVEN**. The table displays the average results of ten trials.

| Models | Average | Center | L-R | U-D | O-IC | O-IG | 2×2Grid | 3×3Grid |
|---|---|---|---|---|---|---|---|---|
| GCA-I | 12.0/24.1 | 14.0/30.2 | 7.9/22.4 | 7.5/26.9 | 13.4/32.9 | 15.5/25.0 | 11.3/16.3 | 14.5/15.3 |
| GCA-R | 13.8/27.4 | 16.6/34.5 | 9.4/26.9 | 6.9/28.0 | 17.3/37.8 | 16.7/26.0 | 11.7/19.2 | 18.1/19.3 |
| GCA-C | 32.7/41.7 | 37.3/51.8 | 26.4/44.6 | 21.5/42.6 | 30.2/46.7 | 33.0/35.6 | 37.6/38.1 | 43.0/32.4 |
| ALANS | 54.3/62.8 | 42.7/63.9 | 42.4/60.6 | 46.2/65.6 | 49.5/64.8 | 53.6/52.0 | 70.5/66.4 | 75.1/65.7 |
| PrAE | 80.0/85.7 | 97.3/**99.9** | 96.2/97.9 | 96.7/97.7 | 95.8/98.4 | 68.6/76.5 | **82.0/84.5** | 23.2/45.1 |
| LGPP | 6.4/16.3 | 9.2/20.1 | 4.7/18.9 | 5.2/21.2 | 4.0/13.9 | 3.1/12.3 | 8.6/13.7 | 10.4/13.9 |
| ANP | 7.3/27.6 | 9.8/47.4 | 4.1/20.3 | 3.5/20.7 | 5.4/38.2 | 7.6/36.1 | 10.0/15.0 | 10.5/15.6 |
| CLAP | 17.5/32.8 | 30.4/42.9 | 13.4/35.1 | 12.2/32.1 | 16.4/37.5 | 9.5/26.0 | 16.0/20.1 | 24.3/35.8 |
| Transformer | 40.1/64.0 | 98.4/99.2 | 67.0/91.1 | 60.9/86.6 | 14.5/69.9 | 13.5/57.1 | 14.7/25.2 | 11.6/18.6 |
| RAISE | **90.0/92.1** | **99.2**/99.8 | **98.5/99.6** | **99.3/99.9** | **97.6/99.6** | **89.3/96.0** | 68.2/71.3 | **77.7/78.7** |

where $\beta_1$, $\beta_2$, and $\beta_3$ are hyperparameters. RAISE also supports semi-supervised training settings. For samples that do not provide rule annotations, RAISE can set $\beta_3 = 0$ and update parameters via the unsupervised part $\mathcal{L}_{\mathrm{rec}} - \beta_1 \mathcal{R}_{\mathrm{pred}} - \beta_2 \mathcal{R}_{\mathrm{rule}}$.

## 4 EXPERIMENTS

In the experiments, we compare the performance of RAISE with other generative solvers by generating answers at the bottom right and, more challenging, arbitrary positions. Then we conduct experiments to visualize the latent concepts learned from the dataset. Finally, RAISE carries out the odd-one-out task and is tested in held-out configurations to illustrate the benefit of learning latent concepts and atomic rules in generative abstract reasoning.

**Datasets.** The models in the experiments are evaluated on the RAVEN (Zhang et al., 2019a) and I-RAVEN (Hu et al., 2021) datasets having seven image configurations (e.g., scenes with centric objects or object grids) and four basic rules. I-RAVEN follows the same configurations as RAVEN and reduces the bias of candidate sets to resist the shortcut learning of models (Hu et al., 2021). See Appendix B for details of datasets.

**Compared Models.** In the task of bottom-right answer selection, we compare RAISE with the powerful generative solvers ALANS (Zhang et al., 2021b), PrAE (Zhang et al., 2021a), and the model proposed by Niv et al. (called GCA for convenience) (Pekar et al., 2020). RAISE selects the candidate closest to the predicted result in the latent space as the answer. We apply three strategies of answer selection in GCA: selecting the candidate having the smallest pixel difference to the prediction (GCA-I), having the smallest difference in the representation space (GCA-R), and having the highest panel score (GCA-C). Since these generative solvers cannot generate non-bottom-right answers, we take Transformer (Vaswani et al., 2017), ANP (Kim et al., 2019), LGPP (Shi et al., 2021), and CLAP (Shi et al., 2023) as baseline models to evaluate the ability to generate answers at arbitrary positions. We provide more details in Appendix C.

**Training and Evaluation Settings.** For non-grid layouts, RAISE is trained under semi-supervised settings by using 5% rule annotations. RAISE leverages 20% rule annotations on O-IG and full rule annotations on 2×2Grid and 3×3Grid. The powerful generative solvers use full rule annotations and are trained and tested on each configuration respectively. We compare RAISE with them to illustrate the acquired bottom-right answer selection ability of RAISE under semi-supervised settings. The baselines can generate answers at arbitrary positions but cannot leverage rule annotations since they do not explicitly model the category of rules. We compare RAISE with the baselines to illustrate the benefit of learning latent concepts and atomic rules for generative abstract reasoning. Since the training of RAISE and the baselines do not require the candidate sets, and RAVEN/I-RAVEN only differ in the distribution of candidates, we train RAISE and the baselines on RAVEN and test them on RAVEN/I-RAVEN directly. See Appendix C for detailed training and evaluation settings.

### 4.1 BOTTOM-RIGHT ANSWER SELECTION

This experiment conducts classical RPM tests that require models to find the missing bottom-right images in eight candidates. Table 1 illustrates RAISE's outstanding generative abstract reason-

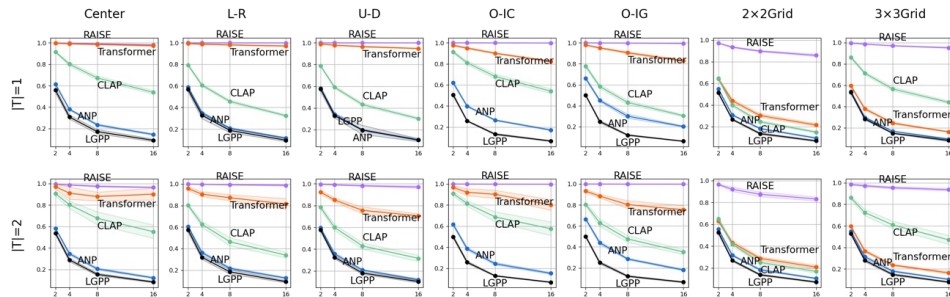

Figure 2: **Selection accuracy at arbitrary positions**. The selection accuracy of RAISE (purple), Transformer (orange), CLAP (green), ANP (blue), and LGPP (black) in arbitrary positions. The x-axis of each plot indicates the number of candidates, and the y-axis is the selection accuracy.

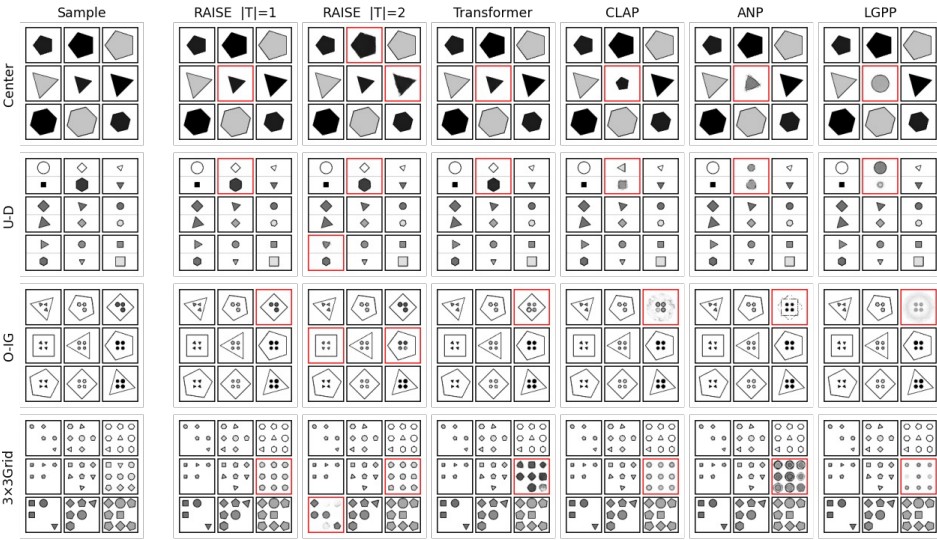

Figure 3: **Answer generation at arbitrary positions**. The prediction results on RAVEN are highlighted (red box) to illustrate the arbitrary-position generation ability. Due to the existence of noise, some predictions may differ from the original sample, but they still follow the correct rules.

ing ability on RAVEN/I-RAVEN. By comparing the difference between predictions and candidates, RAISE outperforms the compared generative solvers in most configurations of RAVEN/I-RAVEN, even if the distractors in candidate sets are not used in training. All the powerful generative solvers take full rule annotations for training, while RAISE in non-grid configurations only requires a small amount of rule annotations (5% samples) to achieve high selection accuracy. RAISE attains the highest selection accuracy compared to the baselines which can generate answers at arbitrary positions. By comparing the results on RAVEN/I-RAVEN, we find that generative solvers are more likely to have accuracy improvement on I-RAVEN, because I-RAVEN generates distractors that are less similar to correct answers to avoid significant biases in candidate sets. For grid-shaped configurations, we found that the noise in datasets will significantly influence the model performance. By removing the noise in object attributes, RAISE achieves high selection accuracy on three grid-shaped configurations using only 20% rule annotations. See Appendix D.1 for the detailed experimental results.

## 4.2 ANSWER SELECTION AT ARBITRARY POSITIONS

The above generative solvers can hardly generate answers at non-bottom-right positions. In this experiment, we probe the ability of RAISE and baselines to generate answers at arbitrary positions. We first generate additional candidate sets in the experiment because RAVEN and I-RAVEN do not provide candidate sets for non-bottom-right images. To this end, we sample a batch of RPMs from the dataset and split the RPMs into target and context images in the same way. For each matrix, we

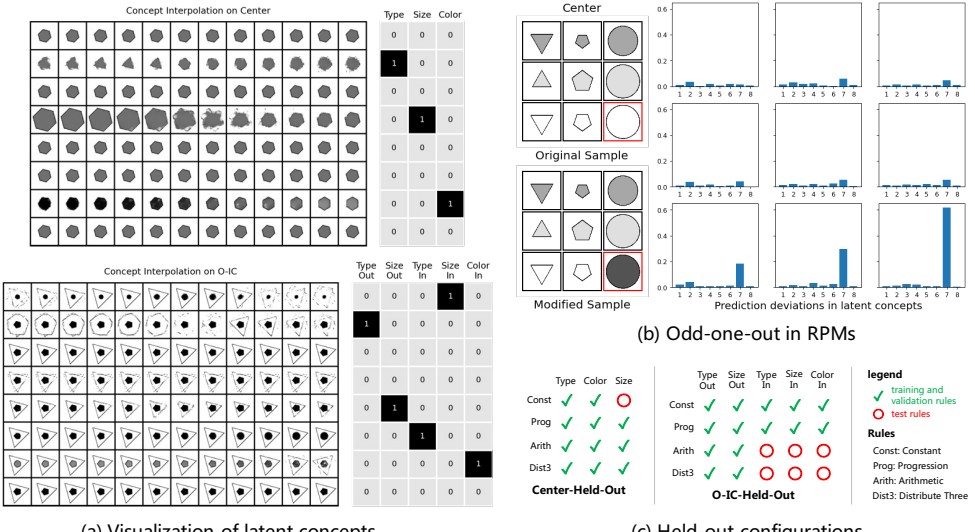

Figure 4: Panel (a) shows the interpolation results of latent concepts and the correspondence between the concepts and attributes. Panel (b) provides an example of RPM-based odd-one-out tests and displays the prediction deviations in concepts of each image. Panel (c) illustrates the strategy to split rule-attribute combinations in held-out configurations.

use the target images of other $N_c$ samples in the batch as distractors to generate a candidate set with $N_c+1$ entries. This strategy can adapt to the missing images at arbitrary and even multiple positions, and we can easily control the difficulty of answer selection through the number of distractors

Figure 2 displays the accuracy of RAISE and baselines when generating answers at arbitrary and multiple positions. RAISE maintains high accuracy in all configurations. Although Transformer has higher accuracy than the other three baselines, especially in non-grid scenes, the prediction accuracy drops significantly on $2\times2$Grid and $3\times3$Grid. Figure 3 provides the qualitative prediction results on RAVEN. It is difficult for ANP and LGPP to generate clear answers. CLAP can generate answers with partially correct attributes in simple cases (e.g., CLAP generates an object with the correct color but the wrong size and shape in the sample of Center). RAISE produces high-quality predictions and can solve RPMs with multiple missing images. By predicting multiple missing images at arbitrary positions, The qualitative results intuitively reveal the in-depth generative abstract reasoning ability in models, which the bottom-right answer generation task does not involve.

## 4.3 LATENT CONCEPTS

Latent concepts bridge atomic rules and high-dimensional observations. Figure 4a visualizes the latent concepts learned from Center and O-IC by traversing concept representations of an image in the latent space. If the concepts are well decomposed, decoding the interpolated concept representations will change one attribute of the original image. Besides observing visualization results, we can find the correspondence between concepts and attributes with the aid of the binary matrix $M$. As shown in Figure 4a, RAISE can automatically set some redundant concepts when there are more concepts than attributes. (e.g., the first concept of *Center*). The visualization results illustrate the concept learning ability of RAISE, which is the foundation of abstracting and selecting atomic rules shared among RPMs.

## 4.4 ODD-ONE-OUT IN RPM

In odd-one-out tests, RAISE attempts to find the rule-breaking image in a panel. To generate RPM-based odd-one-out problems, we replace the bottom-right image of an RPM with a random distractor in the candidate set. Taking Figure 4b as an example, we change the object color from white to black by replacing the bottom-right image. RAISE takes each image in an RPM as the target, gets the prediction results, and computes the prediction error on latent concepts. The right panel of

Table 2: Selection accuracy (%) on two held-out configurations.

| OOD Settings | RAISE | PrAE | ALANS | GCA-C | GCA-R | GCA-I | Transformer | ANP | LGPP | CLAP-NP |
|---|---|---|---|---|---|---|---|---|---|---|
| Center-Held-Out | 99.2 | **99.8** | 46.9 | 35.0 | 14.4 | 12.1 | 12.1 | 10.6 | 8.6 | 19.5 |
| O-IC-Held-Out | **56.1** | 40.5 | 33.4 | 10.1 | 5.3 | 4.9 | 15.8 | 7.5 | 4.6 | 8.6 |

Figure 4b shows the concept-level prediction errors, and we find that the 7th concept of the bottom-right image deviates the most. According to Figure 4a, the 7th concept on *Center* represents the attribute *Color*, which is indeed the attribute modified when constructing the test. The last row has relatively higher concept distances since the incorrect image tends to influence the accuracy of answer generation at the most related positions. Because of the independent latent concepts and concept-specific reasoning processes of RAISE, the high concept distances only appear in the 7th concept. By solving RPM-based odd-one-out problems, we explain how concept-level predictions improve the interpretability of answer selection. Although RAISE is tasked with generating answers, it can handle answer-selection problems by excluding candidates violating the underlying rules.

## 4.5 HELD-OUT CONFIGURATIONS

To explore the abstract reasoning ability on out-of-distribution (OOD) samples, we construct two held-out configurations based on RAVEN (Barrett et al., 2018) as illustrated in Figure 4c. (1) **Center-Held-Out** keeps the samples of *Center* following the attribute-rule tuple *(Size, Constant)* as test samples, and the remaining constitute the training and validation sets. (2) **O-IC-Held-Out** keeps the samples of *O-IC* following the attribute-rule tuples *(Type In, Arithmetic)*, *(Size In, Arithmetic)*, *(Color In, Arithmetic)*, *(Type In, Distribute Three)*, *(Size In, Distribute Three)*, and *(Color In, Distribute Three)* as test samples. The results given in Table 2 indicate that RAISE maintains relatively higher selection accuracy when encountering unseen combinations of attributes and rules. RAISE learns interpretable latent concepts to conduct concept-specific reasoning, by which the learning of rules and concepts are decoupled. Thus RAISE can tackle OOD samples via compositional generalization. Although RAISE has not ever seen the attribute-rule tuple *(Size, Constant)* in training, it can still apply the atomic rule *Constant* learned from other attributes to *Size* in the test phase.

## 5 CONCLUSION AND DISCUSSION

This paper proposes a generative RPM solver RAISE based on conditional deep latent variable models. RAISE can abstract atomic rules from PRMs, keep them in the global knowledge set, and predict target images by selecting proper rules. As the foundation of rule abstraction and selection, RAISE learns interpretable latent concepts from images to decompose the integrated rules of RPMs into atomic rules. Qualitative and quantitative experiments show that RAISE can generate answers at arbitrary positions and outperform baselines, showing outstanding generative abstract reasoning. The odd-one-out task and held-out configurations verify the interpretability of RAISE in concept learning and rule abstraction. By using prediction deviations on concepts, RAISE can find the position and concept that breaks the rules in odd-one-out tasks. By combining the learned latent concepts and atomic rules, RAISE can generate answers on samples with unseen attribute-rule tuples.

**Limitations and Discussion.** The noise in data is a challenge for the models based on conditional generation. In the experiment, we find that the noise of object attributes in grids will influence the selection accuracy of generative solvers like RAISE and Transformer on *2×2Grid*. The candidate sets can provide clearer supervision in training to reduce the impact of noise. Deep latent variable models (DLVMs) can potentially handle noise in RPMs since RAISE works well on *Center* and *O-IC* with noisy attributes like *Rotation*. In future works, exploring appropriate ways to reduce the influence of noise is the key to realizing generative abstract reasoning in more complicated scenes. For generative solvers that do not rely on candidate sets or are completely unsupervised, whether using datasets with large amounts of noise benefits the acquisition of generative abstract reasoning ability is worth exploring since the noise can make a generative problem have numerous solutions (e.g., PGM (Barrett et al., 2018)). In Appendices B.2 and D.1, we conduct an initial experiment and discussion on the impact of noise, but a more systematic and in-depth study will be carried out in the follow-up works. Some recent neural approaches attempt to solve similar systematic generalization problems (Rahaman et al., 2021; Lake & Baroni, 2023). We provide a discussion on the Bayesian and neural approaches of concept learning in Appendix E.

ACKNOWLEDGMENTS

This work was supported by the National Natural Science Foundation of China (No.62176060) and the Program for Professor of Special Appointment (Eastern Scholar) at Shanghai Institutions of Higher Learning.

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

## A    PROOFS AND DERIVATIONS

### A.1    REFORMULATION OF THE POSTERIOR DISTRIBUTION

According to Bayes' theorem, the posterior distribution of rule indicators $q(r^c|\mathbf{z}_S^c, \mathbf{z}_T^c)$ is the product of the conditional prior $q(r^c|\mathbf{z}_S^c)$ and the likelihood $p(\mathbf{z}_T^c|r^c, \mathbf{z}_S^c)$:

$$q(r^c|\mathbf{z}_S^c, \mathbf{z}_T^c) = \frac{p(\mathbf{z}_T^c|r^c, \mathbf{z}_S^c)p(r^c|\mathbf{z}_S^c)}{\sum_{k=1}^{K} p(\mathbf{z}_T^c|r^c = k, \mathbf{z}_S^c)p(r^c = k|\mathbf{z}_S^c)} \propto p(\mathbf{z}_T^c|r^c, \mathbf{z}_S^c)p(r^c|\mathbf{z}_S^c). \quad (13)$$

Considering that $p(\mathbf{z}_T^c|r^c, \mathbf{z}_S^c)$ is an isotropic Gaussian $\mathcal{N}\left(h\left(\mathbf{Z}^c; \psi_{r^c}\right), \sigma_z^2\mathbf{I}\right)$, Equation 13 becomes

$$q(r^c|\mathbf{z}_S^c, \mathbf{z}_T^c) \propto \frac{1}{\sqrt{2\pi\sigma_z^{\mathrm{D}(\mathbf{z}_T^c)}}} \exp\left(-\frac{1}{2\sigma_z^2}\left\|\mathbf{z}_T^c - h\left(\mathbf{Z}^c; \psi_{r^c}\right)\right\|_2^2\right) p(r^c|\mathbf{z}_S^c)$$

$$\propto \exp\left(-\frac{1}{2\sigma_z^2}\left\|\mathbf{z}_T^c - h\left(\mathbf{Z}^c; \psi_{r^c}\right)\right\|_2^2\right) p(r^c|\mathbf{z}_S^c), \quad (14)$$

where $\mathrm{D}(\mathbf{z}_T^c)$ is the size of $\mathbf{z}_T^c$. In practice, RAISE predicts unnormalized logits $\tilde{\boldsymbol{l}}_{1:K}^c$ instead of the probabilities $\tilde{\boldsymbol{\pi}}_{1:K}^c$. Therefore, we use the logarithmic version of Equation 14:

$$\log q(r^c|\mathbf{z}_S^c, \mathbf{z}_T^c) = -\frac{1}{2\sigma_z^2}\left\|\mathbf{z}_T^c - h\left(\mathbf{Z}^c; \Psi_{r^c}\right)\right\|_2^2 + \log p(r^c|\mathbf{z}_S^c) + C\left(\mathbf{z}_S^c, \mathbf{z}_T^c\right). \quad (15)$$

Since the constant $C(\mathbf{z}_S^c, \mathbf{z}_T^c)$ in Equation 15 will not influence the results of normalization, RAISE ignores the constant term and predicts the unnormalized logits via

$$\tilde{l}_k^c = -\frac{1}{2\sigma_z^2}\left\|\mathbf{z}_T^c - h\left(\mathbf{Z}^c; \Psi_k\right)\right\|_2^2 + \log p(r^c = k|\mathbf{z}_S^c)$$

$$= -\frac{1}{2\sigma_z^2}\left\|\mathbf{z}_T^c - h\left(\mathbf{Z}^c; \Psi_k\right)\right\|_2^2 + \log \pi_k^c, \quad k = 1, ..., K. \quad (16)$$

Finally, the variational distribution $q(r^c|\mathbf{z}_S^c, \mathbf{z}_T^c)$ is parameterized by

$$q(r^c|\mathbf{z}_S^c, \mathbf{z}_T^c) = \mathrm{Categorical}\left(\tilde{\boldsymbol{\pi}}_{1:K}^c\right), \quad \text{where } \tilde{\pi}_k^c = \frac{\exp\left(\tilde{l}_k^c\right)}{\sum_{k=1}^{K} \exp\left(\tilde{l}_k^c\right)} \text{ for } k = 1, ..., K. \quad (17)$$

### A.2    DERIVATION OF THE ELBO

With the variational distribution $q_\Psi(\mathbf{h}|\mathbf{x}_T, \mathbf{x}_S)$, the ELBO $\mathcal{L}$ is (Sohn et al., 2015)

$$\log p_\Theta\left(\mathbf{x}_T|\mathbf{x}_S\right) \geq \mathbb{E}_{q_\Psi(\mathbf{h}|\mathbf{x}_T, \mathbf{x}_S)}\left[\log \frac{p_\Theta\left(\mathbf{h}, \mathbf{x}_T|\mathbf{x}_S\right)}{q_\Psi(\mathbf{h}|\mathbf{x}_T, \mathbf{x}_S)}\right] = \mathcal{L}. \quad (18)$$

Considering the generative and inference processes

$$p_\Theta(\mathbf{h}, \mathbf{x}_T|\mathbf{x}_S) = \prod_{t\in T} p_\varphi(\mathbf{x}_t|\mathbf{z}_t)\prod_{c=1}^{C}\left(p_\psi(\mathbf{z}_T^c|r^c, \mathbf{z}_S^c)p_\phi(r^c|\mathbf{z}_S^c)\prod_{s\in S} p_\theta(\mathbf{z}_s^c|\mathbf{x}_s)\right),$$

$$q_\Psi(\mathbf{h}|\mathbf{x}_T, \mathbf{x}_S) = \prod_{c=1}^{C}\left(q_{\phi,\psi}(r^c|\mathbf{z}_S^c, \mathbf{z}_T^c)\prod_{s\in S} q_\theta(\mathbf{z}_s^c|\mathbf{x}_s)\prod_{t\in T} q_\theta(\mathbf{z}_t^c|\mathbf{x}_t)\right), \quad (19)$$

Equation 18 is further decomposed by

$$
\begin{aligned}
\mathcal{L} &= \mathbb{E}_{q_\Psi(\boldsymbol{h}|\boldsymbol{x}_T,\boldsymbol{x}_S)}\left[\log\prod_{t\in T}p_\varphi(\boldsymbol{x}_t|\boldsymbol{z}_t)\right] \\
&\quad - \mathbb{E}_{q_\Psi(\boldsymbol{h}|\boldsymbol{x}_T,\boldsymbol{x}_S)}\left[\log\prod_{c=1}^{C}\frac{q_{\phi,\psi}(r^c|\boldsymbol{z}_S^c,\boldsymbol{z}_T^c)\prod_{s\in S}q_\theta(\boldsymbol{z}_s^c|\boldsymbol{x}_s)\prod_{t\in T}q_\theta(\boldsymbol{z}_t^c|\boldsymbol{x}_t)}{p_\psi(\boldsymbol{z}_T^c|r^c,\boldsymbol{z}_S^c)p_\phi(r^c|\boldsymbol{z}_S^c)\prod_{s\in S}p_\theta(\boldsymbol{z}_s^c|\boldsymbol{x}_s)}\right] \\
&= \sum_{t\in T}\mathbb{E}_{q_\Psi(\boldsymbol{h}|\boldsymbol{x}_T,\boldsymbol{x}_S)}\left[\log p_\varphi(\boldsymbol{x}_t|\boldsymbol{z}_t)\right] - \sum_{c=1}^{C}\mathbb{E}_{q_\Psi(\boldsymbol{h}|\boldsymbol{x}_T,\boldsymbol{x}_S)}\left[\log\frac{q_\theta(\boldsymbol{z}_T^c|\boldsymbol{x}_T)}{p_\psi(\boldsymbol{z}_T^c|r^c,\boldsymbol{z}_S^c)}\right] \\
&\quad - \sum_{c=1}^{C}\mathbb{E}_{q_\Psi(\boldsymbol{h}|\boldsymbol{x}_T,\boldsymbol{x}_S)}\left[\log\frac{q_{\phi,\psi}(r^c|\boldsymbol{z}_S^c,\boldsymbol{z}_T^c)}{p_\phi(r^c|\boldsymbol{z}_S^c)}\right] - \sum_{c=1}^{C}\sum_{s\in S}\mathbb{E}_{q_\Psi(\boldsymbol{h}|\boldsymbol{x}_T,\boldsymbol{x}_S)}\left[\log\frac{q_\theta(\boldsymbol{z}_s^c|\boldsymbol{x}_s)}{p_\theta(\boldsymbol{z}_s^c|\boldsymbol{x}_s)}\right].
\end{aligned}
\tag{20}
$$

Since the encoder is shared between the generative and inference processes, we have $q_\theta(\boldsymbol{z}_s^c|\boldsymbol{x}_s) = p_\theta(\boldsymbol{z}_s^c|\boldsymbol{x}_s)$ and

$$
\sum_{c=1}^{C}\sum_{s\in S}\mathbb{E}_{q_\Psi(\boldsymbol{h}|\boldsymbol{x}_T,\boldsymbol{x}_S)}\left[\log\frac{q_\theta(\boldsymbol{z}_s^c|\boldsymbol{x}_s)}{p_\theta(\boldsymbol{z}_s^c|\boldsymbol{x}_s)}\right] = 0.
\tag{21}
$$

Therefore, the ELBO is

$$
\begin{aligned}
\mathcal{L} &= \underbrace{\sum_{t\in T}\mathbb{E}_{q_\Psi(\boldsymbol{h}|\boldsymbol{x}_T,\boldsymbol{x}_S)}\left[\log p_\varphi(\boldsymbol{x}_t|\boldsymbol{z}_t)\right]}_{\mathcal{L}_{\text{rec}}} - \underbrace{\sum_{c=1}^{C}\mathbb{E}_{q_\Psi(\boldsymbol{h}|\boldsymbol{x}_T,\boldsymbol{x}_S)}\left[\log\frac{q_\theta(\boldsymbol{z}_T^c|\boldsymbol{x}_T)}{p_\psi(\boldsymbol{z}_T^c|r^c,\boldsymbol{z}_S^c)}\right]}_{\mathcal{R}_{\text{pred}}} \\
&\quad - \underbrace{\sum_{c=1}^{C}\mathbb{E}_{q_\Psi(\boldsymbol{h}|\boldsymbol{x}_T,\boldsymbol{x}_S)}\left[\log\frac{q_{\phi,\psi}(r^c|\boldsymbol{z}_S^c,\boldsymbol{z}_T^c)}{p_\phi(r^c|\boldsymbol{z}_S^c)}\right]}_{\mathcal{R}_{\text{rule}}}.
\end{aligned}
\tag{22}
$$

## A.3 Monte Carlo Estimator of the ELBO

For a given RPM problem $(\boldsymbol{x}_S,\boldsymbol{x}_T)$, we sample the latent variables $\tilde{r}$, $\tilde{\boldsymbol{z}}_S$, and $\tilde{\boldsymbol{z}}_T$ from the variatonal posterior $q_\Psi(\boldsymbol{h}|\boldsymbol{x}_T,\boldsymbol{x}_S)$ to compute the ELBO:

$$
\begin{aligned}
\tilde{\boldsymbol{z}}_s^c &\sim \mathcal{N}\left(\tilde{\boldsymbol{\mu}}_s^c,\sigma_z^2\boldsymbol{I}\right), & s\in S, \quad c=1,...,C, \\
\tilde{\boldsymbol{z}}_t^c &\sim \mathcal{N}\left(\tilde{\boldsymbol{\mu}}_t^c,\sigma_z^2\boldsymbol{I}\right), & t\in T, \quad c=1,...,C, \\
\tilde{r}^c &\sim \text{Categorical}\left(\tilde{\boldsymbol{\pi}}_{1:K}^c\right), & c=1,...,C.
\end{aligned}
\tag{23}
$$

$\tilde{\boldsymbol{\mu}}_s^{1:C} = g_\theta^{\text{enc}}(\boldsymbol{x}_s)$ and $\tilde{\boldsymbol{\mu}}_t^{1:C} = g_\theta^{\text{enc}}(\boldsymbol{x}_t)$ are means of latent concepts computed by the encoder. $\tilde{\boldsymbol{\pi}}_{1:K}^c$ is given by 17 and the indicator $\tilde{r}^c$ is sampled through the Gumbel-Softmax distribution (Jang et al., 2016). Using the Monte Carlo estimator, $\mathcal{L}$ can be approximated by the sampled latent variables.

### A.3.1 Reconstruction Loss

$$
\mathcal{L}_{\text{rec}} \approx \sum_{t\in T}\log p_\varphi(\boldsymbol{x}_t|\tilde{\boldsymbol{z}}_t) = -\frac{1}{2\sigma_x^2}\sum_{t\in T}\left\|\boldsymbol{x}_t-\tilde{\boldsymbol{\Lambda}}_t\right\|_2^2 + C_{\text{rec}}, \quad \text{where } \tilde{\boldsymbol{\Lambda}}_t = g_\varphi^{\text{dec}}(\tilde{\boldsymbol{z}}_t^{1:C})
\tag{24}
$$

### A.3.2 CONCEPT REGULARIZER

$$
\begin{aligned}
\mathcal{R}_{\text{pred}} &= \sum_{c=1}^{C} \mathbb{E}_{q_\theta(\boldsymbol{z}_T^c|\boldsymbol{x}_T)} \left[ \mathbb{E}_{q_\theta(\boldsymbol{z}_S^c|\boldsymbol{x}_S)} \left[ \mathbb{E}_{q_{\phi,\psi}(r^c|\boldsymbol{z}_S^c, \boldsymbol{z}_T^c)} \left[ \log \frac{q_\theta(\boldsymbol{z}_T^c|\boldsymbol{x}_T)}{p_\psi(\boldsymbol{z}_T^c|r^c, \boldsymbol{z}_S^c)} \right] \right] \right] \\
&\approx \sum_{c=1}^{C} \mathbb{E}_{q_\theta(\boldsymbol{z}_S^c|\boldsymbol{x}_S)} \left[ \mathbb{E}_{p_\star(r^c|\boldsymbol{x})} \left[ \mathbb{E}_{q_\theta(\boldsymbol{z}_T^c|\boldsymbol{x}_T)} \left[ \log \frac{q_\theta(\boldsymbol{z}_T^c|\boldsymbol{x}_T)}{p_\psi(\boldsymbol{z}_T^c|r^c, \boldsymbol{z}_S^c)} \right] \right] \right] \\
&= \sum_{c=1}^{C} \mathbb{E}_{q_\theta(\boldsymbol{z}_S^c|\boldsymbol{x}_S)} \left[ \mathbb{E}_{p_\star(r^c|\boldsymbol{x})} \left[ D_{\text{KL}} \left( q_\theta(\boldsymbol{z}_T^c|\boldsymbol{x}_T) \| p_\psi(\boldsymbol{z}_T^c|r^c, \boldsymbol{z}_S^c) \right) \right] \right] \\
&\approx \sum_{c=1}^{C} D_{\text{KL}} \left( q_\theta(\boldsymbol{z}_T^c|\boldsymbol{x}_T) \big\| p_\psi(\boldsymbol{z}_T^c|\tilde{r}^c, \tilde{\boldsymbol{z}}_S^c) \right) = \frac{1}{2\sigma_z^2} \sum_{c=1}^{C} \left\| \tilde{\boldsymbol{\mu}}_T^c - \boldsymbol{\mu}_T^c \right\|_2^2 + C_{\text{pred}}.
\end{aligned}
\tag{25}
$$

Trained with rule annotations, RAISE can quickly approach the real distribution $p_\star(r^c|\boldsymbol{x})$ provided in rule annotations after the early learning stage. Therefore, we regard the real distribution as the predicted rule distribution, which is related to the matrix rather than conditional on the latent concepts. That is, we assume that samples from $p_\star(r^c|\boldsymbol{x})$ are similarly distributed to those from $q_{\phi,\psi}(r^c|\tilde{\boldsymbol{z}}_S^c, \tilde{\boldsymbol{z}}_T^c)$ after a few learning epochs. By replacing the $q_{\phi,\psi}(r^c|\tilde{\boldsymbol{z}}_S^c, \tilde{\boldsymbol{z}}_T^c)$ to $p_\star(r^c|\boldsymbol{x})$, we move the inner expectation on $r^c$ to the front. In this way, the inner expectation becomes the KL divergence between Gaussians and has a closed-form solution, reducing the additional noise in the sampling process.

### A.3.3 RULE REGULARIZER

$$
\begin{aligned}
\mathcal{R}_{\text{rule}} &= \sum_{c=1}^{C} \mathbb{E}_{q_\theta(\boldsymbol{z}_T^c|\boldsymbol{x}_T)} \left[ \mathbb{E}_{q_\theta(\boldsymbol{z}_S^c|\boldsymbol{x}_S)} \left[ \mathbb{E}_{q_{\phi,\psi}(r^c|\boldsymbol{z}_S^c, \boldsymbol{z}_T^c)} \left[ \log \frac{q_{\phi,\psi}(r^c|\boldsymbol{z}_S^c, \boldsymbol{z}_T^c)}{p_\phi(r^c|\boldsymbol{z}_S^c)} \right] \right] \right] \\
&= \sum_{c=1}^{C} \mathbb{E}_{q_\theta(\boldsymbol{z}_T^c|\boldsymbol{x}_T)} \left[ \mathbb{E}_{q_\theta(\boldsymbol{z}_S^c|\boldsymbol{x}_S)} \left[ D_{\text{KL}} \left( q_{\phi,\psi}(r^c|\boldsymbol{z}_S^c, \boldsymbol{z}_T^c) \big\| p_\phi(r^c|\boldsymbol{z}_S^c) \right) \right] \right] \\
&\approx \sum_{c=1}^{C} D_{\text{KL}} \left( q_{\phi,\psi}(r^c|\tilde{\boldsymbol{z}}_S^c, \tilde{\boldsymbol{z}}_T^c) \big\| p_\phi(r^c|\tilde{\boldsymbol{z}}_S^c) \right) = \sum_{c=1}^{C} \sum_{k=1}^{K} \tilde{\pi}_k^c \log \frac{\tilde{\pi}_k^c}{\pi_k^c}.
\end{aligned}
\tag{26}
$$

### A.3.4 ELBO

Ignoring the constant terms $C_{\text{rec}}$ and $C_{\text{pred}}$, the approximation of ELBO is

$$
\mathcal{L} \approx \underbrace{-\frac{1}{2\sigma_x^2} \sum_{t \in T} \left\| \boldsymbol{x}_t - \tilde{\boldsymbol{\Lambda}}_t \right\|_2^2}_{\mathcal{L}_{\text{rec}}} \underbrace{- \frac{1}{2\sigma_z^2} \sum_{c=1}^{C} \left\| \tilde{\boldsymbol{\mu}}_T^c - \boldsymbol{\mu}_T^c \right\|_2^2}_{\mathcal{R}_{\text{pred}}} \underbrace{- \sum_{c=1}^{C} \sum_{k=1}^{K} \tilde{\pi}_k^c \log \frac{\tilde{\pi}_k^c}{\pi_k^c}}_{\mathcal{R}_{\text{rule}}}.
\tag{27}
$$

## B DATASETS

### B.1 RAVEN AND I-RAVEN

Figure 5 displays seven image configurations of RAVEN (Zhang et al., 2019a). The image attributes include *Number/Position*, *Type*, *Size*, and *Color*, which can follow the rules *Constant*, *Progress*, *Arithmetic*, and *Distribution Three*. Each configuration contains 6000 training samples, 2000 validation samples, and 2000 test samples. RAVEN provides eight candidate images and attribute-level rule annotations for each RPM problem. Previous work pointed out the existence of bias in candidate sets of RAVEN (Hu et al., 2021), which allows models to find shortcuts for answer selection. I-RAVEN uses Attribute Bisection Tree (ABT) to generate candidate sets to resist shortcut learning (Hu et al., 2021). The experiment shows that the models trained with only the candidate sets of I-RAVEN have a selection accuracy close to the random guesses, which evidences the effectiveness of the candidate generation strategy.

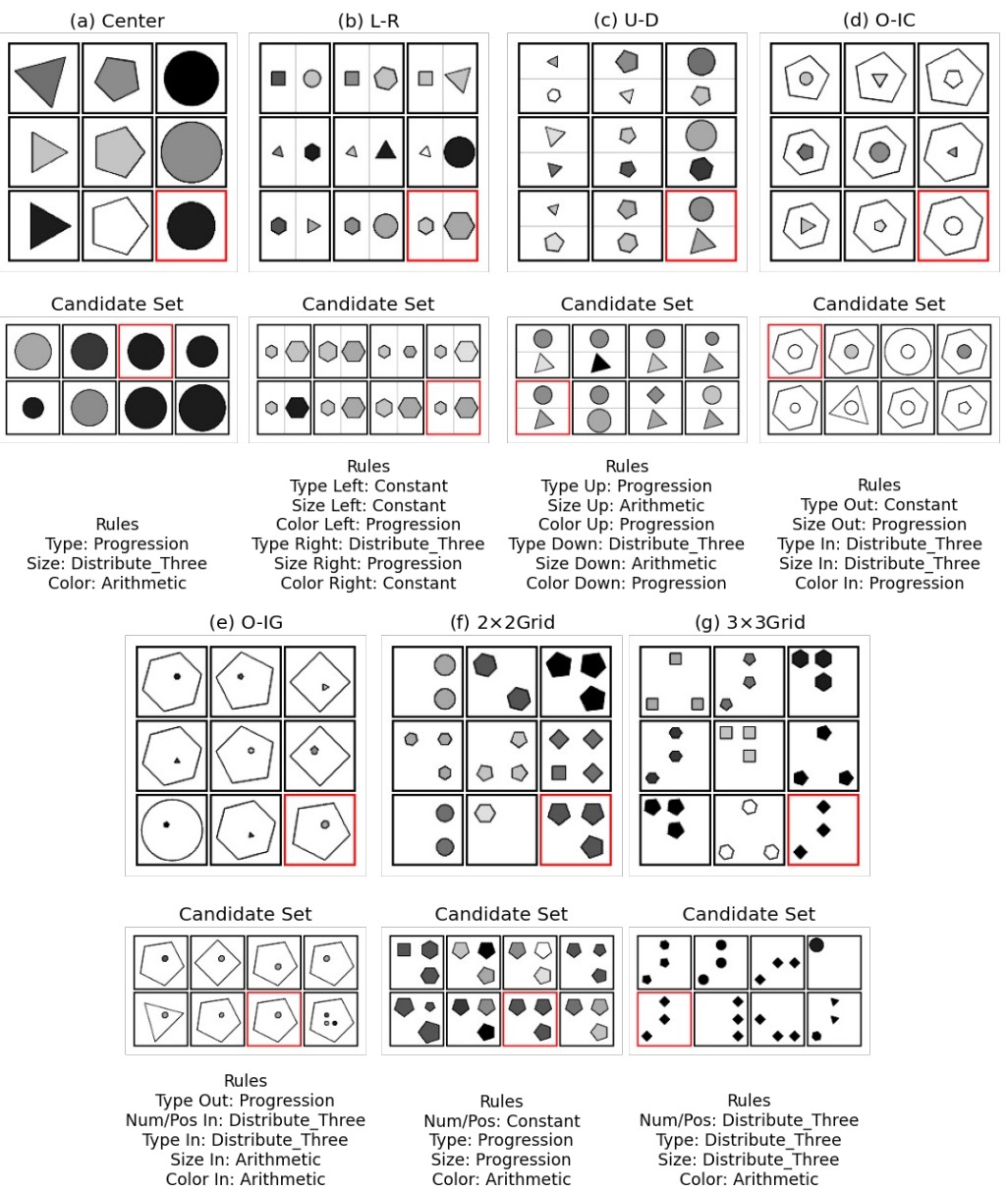

Figure 5: **Different configurations of RAVEN**. In each figure, the top panel is an RPM where the target images are highlighted in red boxes; the middle panel is a candidate set with eight candidate images; and the bottom panel shows the attribute-changing rules in the RPM.

## B.2 ATTRIBUTE NOISE OF RAVEN AND I-RAVEN

RAVEN and I-RAVEN introduce noise to some attributes to increase the complexity of problems. In Center, L-R, U-D, and O-IC, the rotation of objects is the noise attribute. We can keep objects unchanged in rows or make random rotations. Figure 6 displays the noise of object grids on O-IG, 2×2Grid, and 3×3Grid, including the noise of object attributes (i.e., objects in Figure 6c can have different colors and rotations), and the noise of object positions (Figure 6d). The candidate set ensures that only one candidate image is the correct answer. To explore the influence of noise on selection accuracy, we remove the noise of object attributes from object grids, keep the noise of object positions, and generate three configurations O-IG-Uni, 2×2Grid-Uni, and 3×3Grid-Uni.

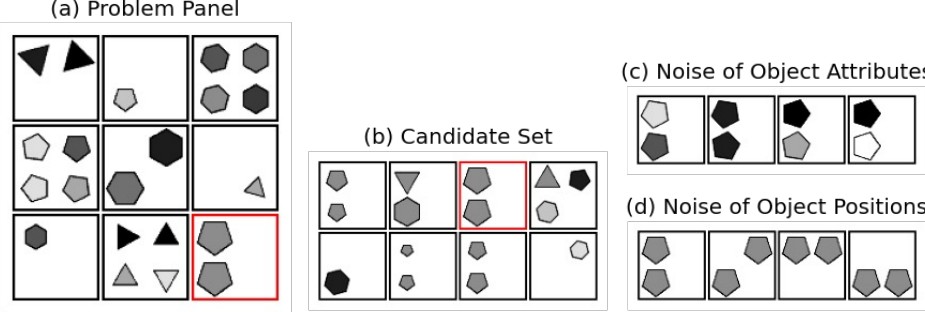

Figure 6: **The illustration of attribute noise**. (a) is an RPM from $2\times2$Grid; (b) is the candidate set; (c) and (d) visualize two possible types of noise in the RPM. In this case, the image is correct as long as there are two pentagons of the correct size. The color, rotation, and position of objects will not influence the correctness of the image.

## C  MODELS

### C.1  RAISE

This section introduces the architectures and hyperparameters of RAISE. The network architectures are introduced in the order of $g_\theta^{\text{enc}}$, $f_{\phi_1}^{\text{row}}$, $f_{\phi_2}^{\text{col}}$, $f_{\phi_3}^{\text{ind}}$, $h$, and $g_\varphi^{\text{dec}}$.

- $g_\theta^{\text{enc}}$. RAISE used a convolutional neural network to downsample images and extract the mean of latent concepts. Denoting the number and size of latent concepts as $C$ and $d_z$, the encoder is
    - $4 \times 4$ Conv, stride 2, padding 1, 64 BatchNorm, ReLU
    - $4 \times 4$ Conv, stride 2, padding 1, 128 BatchNorm, ReLU
    - $4 \times 4$ Conv, stride 2, padding 1, 256 BatchNorm, ReLU
    - $4 \times 4$ Conv, stride 2, padding 1, 512 BatchNorm, ReLU
    - $4 \times 4$ Conv, 512 BatchNorm, ReLU
    - ReshapeBlock, 512
    - Fully Connected, $C \times d_z$

  The ReshapeBlock flattens the feature map of the shape (512, 1, 1) to the vector with 512 dimensions, which is projected and split into the mean of $C$ latent concepts.

- $f_{\phi_1}^{\text{row}}$ and $f_{\phi_2}^{\text{col}}$. The two networks have the same architecture to extract the row and column representations from RPMs:
    - Fully Connected, 512 ReLU
    - Fully Connected, 512 ReLU
    - Fully Connected, 64

  where the input size is $3 \times d_z$ and the size of output row and column representations is 64.

- $f_{\phi_3}^{\text{ind}}$. This network converts the overall row and column representations of an RPM to the logits of selection probabilities for atomic rule selection:
    - Fully Connected, 64 ReLU
    - Fully Connected, 64 ReLU
    - Fully Connected, $K$

  where $K$ is the number of atomic rules. Since the row and column representations are concatenated as the input, the input size of the network is 128.

- $h(\boldsymbol{Z}^c; \psi_k)$. This network is a fully convolutional network, which predicts the means of target latent concepts from the representation matrix $\boldsymbol{Z}^c$:
    - $3 \times 3$ Conv, stride 1, padding 1, 128 ReLU
    - $3 \times 3$ Conv, stride 1, padding 1, 128 ReLU

- $3 \times 3$ Conv, stride 1, padding 1, $d_z$

$h$ adopts convolutional layers with $3\times3$ kernels, stride 1, and padding 1 to keep the shape of the $3\times3$ representation matrix. The global knowledge set $\psi_{1:K}$ stores $K$ learnable parameters of $h$, which represents $K$ atomic rule respectively.

- $g_\varphi^{\text{dec}}$. The decoder accepts all latent concepts of an image as input and outputs the mean of the pixel values for image reconstruction. The architecture is

    - ReshapeBlock, $(C \times d_z, 1, 1)$
    - $1 \times 1$ Deconv, 256 BatchNorm, LeakyReLU
    - $4 \times 4$ Deconv, 128 BatchNorm, LeakyReLU
    - $4 \times 4$ Deconv, stride 2, padding 1, 64 BatchNorm, LeakyReLU
    - $4 \times 4$ Deconv, stride 2, padding 1, 32 BatchNorm, LeakyReLU
    - $4 \times 4$ Deconv, stride 2, padding 1, 32 BatchNorm, LeakyReLU
    - $4 \times 4$ Deconv, stride 2, padding 1, 1 Sigmoid

    where the negative slope of LeakyReLU is 0.02. Since the images of RAVEN and I-RAVEN are grayscaled, the decoder output only one image channel and uses the Sigmoid activation function to scale the range of pixel values to $(0, 1)$.

For all configurations of RAVEN, we set learning rate as $3 \times 10^{-4}$, batch size as 512, $K = 4$, $\sigma_x = 0.1$, $\sigma_z = 0.1$, $C = 8$, $d_z = 8$, $\beta_1 = 5$, $\beta_2 = 20$, and $\beta_3 = 10$. RAISE is insensitive when increasing $C$ since it can generate redundant latent concepts. When $C$ is too small to encode all attributes, the selection accuracy will decline significantly. We can set a large $C$ and reduce it until the number of redundant latent concepts is reasonable. In general, we choose $K$ by directly counting the number of unique labels in rule annotations. RAISE updates the parameters through the RMSprop optimizer (Hinton et al., 2012). To select the best model, we watch the performance on the validation set after each training epoch and save the model with the highest accuracy.

## C.2 POWERFUL GENERATIVE SOLVERS

**ALANS** (Zhang et al., 2021b)  We train ALANS on the codebase released by the authors [2], setting the learning rate as $0.95 \times 10^{-4}$ and the coefficient of the auxiliary loss as 1.0. Since the model can hardly converge from the initialized parameters, we initialize the parameters of ALANS with the pretrained checkpoint provided by the authors. More details can be seen in the repository.

**PrAE** (Zhang et al., 2021a)  For PrAE, we use the commended hyperparameters that the learning rate is $0.95 \times 10^{-4}$ and the weight of auxiliary loss is 1.0. The implementation of PrAE is based on the official repository [3].

**GCA** (Pekar et al., 2020)  The official code of GCA [4] only implements the auxiliary loss on the PGM dataset (Barrett et al., 2018). Therefore, we modify the output size of the auxiliary network to the size of one-hot rule annotations in RAVEN/I-RAVEN. We set the latent size in GCA as 64 and the learning rate as $2 \times 10^{-4}$.

## C.3 BASELINES

**Transformer** (Vaswani et al., 2017)  To improve the model capability, we first apply the encoder and decoder to project images into low-dimensional representations and then predict the targets in the representation space via Transformer. Transformer uses the same encoder and decoder structures as RAISE. The hyperparameters of Transformer are chosen through grid search. We set the learning rate as $1 \times 10^{-4}$ from $\{5 \times 10^{-4}, 1 \times 10^{-4}, 5 \times 10^{-5}\}$, the representation size as 256 from $\{512, 256, 128\}$, and the number of Transformer blocks as 4 from $\{2, 4, 6\}$. In addition, the number of attention heads is 4, the hidden size of feedforward networks is 1024, and the dropout is 0.1. All parameters are updated by the Adam (Kingma & Ba, 2015) optimizer.

---

[2]https://github.com/WellyZhang/ALANS
[3]https://github.com/WellyZhang/PrAE
[4]https://github.com/nivPekar/Generating-Correct-Answers-for-Progressive-Matrices-Intelligence-Tests

Table 3: Learning rates of ANP on RAVEN/I-RAVEN.

| Center | L-R | U-D | O-IC | O-IG | 2×2Grid | 3×3Grid |
|---|---|---|---|---|---|---|
| $5 \times 10^{-5}$ | $1 \times 10^{-5}$ | $1 \times 10^{-5}$ | $5 \times 10^{-6}$ | $5 \times 10^{-6}$ | $3 \times 10^{-5}$ | $3 \times 10^{-5}$ |

Table 4: **Hyperparameters of CLAP**. We give the number of concepts, weights in the ELBO ($\beta_t$, $\beta_f$, and $\beta_{TC}$), and standard deviation $\sigma_z$ on RAVEN/I-RAVEN.

| Hyperparameters | Center | L-R | U-D | O-IC | O-IG | 2×2Grid | 3×3Grid |
|---|---|---|---|---|---|---|---|
| #Concepts | 5 | 10 | 10 | 6 | 8 | 8 | 10 |
| $\beta_t$ | 100 | 50 | 50 | 30 | 30 | 30 | 80 |
| $\beta_f$ | 100 | 50 | 50 | 60 | 30 | 30 | 80 |
| $\beta_{TC}$ | 100 | 50 | 50 | 50 | 30 | 30 | 80 |
| $\sigma_z$ | 0.1 | 0.1 | 0.1 | 0.4 | 0.1 | 0.3 | 0.3 |

**ANP** (Kim et al., 2019)  For all configurations, we set the size of the global latent as $1024$ and the batch size as $512$. Table 3 shows the configuration-specific learning rates. Other hyperparameters and the model architecture remain the same as the 2D regression configuration in the original paper (Kim et al., 2019).

**LGPP** (Shi et al., 2021)  In the experiments, we use the official code of LGPP [5] by setting the learning rate as $5 \times 10^{-4}$ and the batch size as $256$. In terms of model architecture, we set the size of axis latent variables as 4, the size of axis representations as 4, and the input size of the RBF kernel as 8. The network that converts axis latent variables to axis representations has hidden sizes [64, 64]. The network to extract the features for RBF kernels has hidden sizes [128, 128, 128, 128]. The hyperparameter $\beta$ that promotes disentanglement of LGPP is set to 10. For the configuration Center, the number of concepts is 5, while the others use 10 concepts.

**CLAP** (Shi et al., 2023)  Here we adopt the model architecture of the CRPM configuration in the official repository [6] and adjust the learning rate to $5 \times 10^{-4}$, the batch size to 256, and the concept size to 8. Other hyperparameters are displayed in Table 4.

## C.4   COMPUTATIONAL RESOURCE

All the models are trained on the server with Intel(R) Xeon(R) Platinum 8375C CPUs, 24GB NVIDIA GeForce RTX 3090 GPUs, 512GB RAM, and Ubuntu 18.04.6 LTS. RAISE is implemented with PyTorch (Paszke et al., 2019).

## D   ADDITIONAL EXPERIMENTAL RESULTS

### D.1   BOTTOM-RIGHT ANSWER SELECTION

We generate new configurations by removing the noise in object attributes to analyze the influence of noise attributes. As shown in Table 5, RAISE achieves the highest accuracy on all three configurations. When we introduce more noise to RPMs, the number of solutions that follow the correct rules will increase. In this case, the provided candidate set with one correct answer and seven distractors can act as clear supervision in model training. Without the assistance of candidate sets in training, it is challenging to catch rules from noisy RPMs with multiple potential solutions. Therefore, RAISE and Transformer have significant accuracy improvements on configurations with less noise attributes. Overall, the experimental results show that reducing noise can bring significant improvements for the models trained without distractors in candidate sets (such as Transformer and RAISE). RAISE only requires 20% rule annotations to learn atomic rules from low-noise samples.

---

[5]https://github.com/FudanVI/generative-abstract-reasoning/tree/main/rpm-lgpp
[6]https://github.com/FudanVI/generative-abstract-reasoning/tree/main/clap

Table 5: The accuracy (%) of selecting bottom-right answers on O-IG-Uni, 2×2Grid-Uni, and 3×3Grid-Uni.

| Models | O-IG-Uni | 2×2Grid-Uni | 3×3Grid-Uni |
|---|---|---|---|
| GCA-I | 21.2/36.7 | 19.5/23.3 | 20.6/21.6 |
| GCA-R | 20.7/36.3 | 21.9/28.1 | 25.9/25.2 |
| GCA-C | 53.8/37.7 | 58.8/35.6 | 67.0/27.5 |
| PrAE | 29.1/45.1 | 85.4/85.6 | 26.8/47.2 |
| ALANS | 29.7/41.5 | 66.2/55.3 | 84.0/73.3 |
| LGPP | 3.4/12.3 | 4.1/13.0 | 4.0/13.1 |
| ANP | 31.5/34.0 | 10.0/15.6 | 12.0/16.3 |
| CLAP | 14.4/31.7 | 22.5/39.1 | 12.1/32.9 |
| Transformer | 70.6/57.9 | 73.3/73.0 | 34.2/37.0 |
| RAISE | **95.8/99.0** | **87.6/97.9** | **95.3/93.2** |

Table 6: **The accuracy (%) of selecting bottom-right answers on different configurations (i.e., *Center*, *L-R*, etc) of RAVEN/I-RAVEN**. In this table, RAISE is trained without the supervision of rule annotations (-aux) to illustrate the abstract reasoning ability in the unsupervised training setting. The table displays the average results of ten trials.

| Models | Average | Center | L-R | U-D | O-IC | O-IG | 2×2Grid | 3×3Grid |
|---|---|---|---|---|---|---|---|---|
| LGPP | 6.4/16.3 | 9.2/20.1 | 4.7/18.9 | 5.2/21.2 | 4.0/13.9 | 3.1/12.3 | 8.6/13.7 | 10.4/13.9 |
| ANP | 7.3/27.6 | 9.8/47.4 | 4.1/20.3 | 3.5/20.7 | 5.4/38.2 | 7.6/36.1 | 10.0/15.0 | 10.5/15.6 |
| CLAP | 17.5/32.8 | 30.4/42.9 | 13.4/35.1 | 12.2/32.1 | 16.4/37.5 | 9.5/26.0 | 16.0/20.1 | 24.3/**35.8** |
| Transformer | 40.1/64.0 | **98.4/99.2** | **67.0/91.1** | 60.9/86.6 | 14.5/69.9 | 13.5/57.1 | 14.7/25.2 | 11.6/18.6 |
| RAISE (-aux) | **54.5/67.7** | 30.2/56.6 | 47.9/80.8 | **87.0/94.9** | **96.9/99.2** | **56.9/83.9** | **30.4/30.5** | **32.0**/27.8 |

We also provide the selection accuracy of unsupervised RAISE in Table 6. The average accuracy of unsupervised RAISE lies between the unsupervised arbitrary-generation baselines (i.e., LGPP, ANP, CLAP, and Transformer) and the powerful generative RPM solvers trained with full rule annotations (i.e., GCA, ALANS, and PrAE).

## D.2   ANSWER SELECTION AT ARBITRARY POSITION

In this section, we give additional results for arbitrary-position answer generation. Figure 8 provides the detailed results of arbitrary-position answer generation for all seven configurations of RAVEN, for example, the prediction results when $|T| = 1$ (Figure 8a) and $|T| = 2$ (Figure 8b). In the visualization results, RAISE can generate high-quality predictions when $|T| = 1$ and $|T| = 2$. The performance of Transformer varies significantly among different configurations. Transformer predicts accurate answers on Center, while the predictions on 3×3Grid deviate significantly from the ground truth images. In most cases, ANP, LGPP, and CLAP tend to generate incorrect images. Figure 7 provides the selection accuracy on I-RAVEN with different numbers of target images ($|T| = 1, 2$) and different numbers of distractors in candidate sets ($N_c = 1, 3, 7, 15$). We can make further analysis through the selection accuracy with test errors in Tables 8 and 9, where RAISE outperforms other baseline models on all image configurations of RAVEN and I-RAVEN.

## D.3   LATENT CONCEPTS

As mentioned in the main text, concept learning is an important component of RAISE. This section shows the interpolation results of latent concepts on all image configurations and the correspondences between latent concepts and real attributes in Figures 9 and 10. In most configurations, RAISE can learn independent latent concepts and the binary matrix $M$ that accurately reflects the concept-attribute correspondences. RAISE does not assign the latent concepts encoding object rotations to any attribute since the noise attributes are not included in rule annotations. This experiment illustrates the interpretability of the acquired latent concepts, which benefits the prediction of correct answers and the following experiment of odd-one-out.

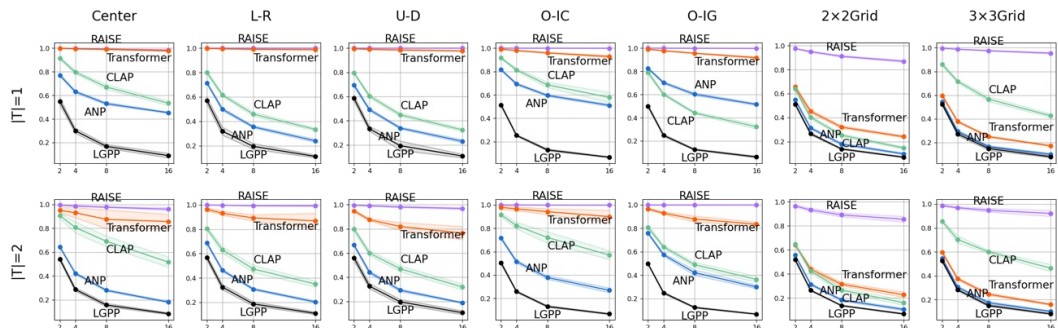

Figure 7: **Selection accuracy at arbitrary positions on I-RAVEN**. Each plot contains the selection accuracy of RAISE (purple), Transformer (orange), CLAP (green), ANP (blue), and LGPP (black). The x-axis is the number of candidates, and the y-axis is the selection accuracy.

Table 7: The accuracy (%) using different strategies of answer selection.

| Models | Average | Center | L-R | U-D | O-IC | O-IG | 2×2Grid | 3×3Grid |
|---|---|---|---|---|---|---|---|---|
| RAISE-latent | **90.0/92.1** | **99.2/99.8** | **98.5/99.6** | **99.3/99.9** | **97.6/99.6** | **89.3/96.0** | **68.2/71.3** | **77.7/78.7** |
| RAISE-pixel | 72.9/77.8 | 95.2/96.8 | 90.6/95.8 | 96.6/98.5 | 80.4/90.6 | 69.1/81.1 | 40.1/42.6 | 38.1/39.5 |

## D.4 ODD-ONE-OUT IN RPM

In this experiment, we provide the additional results of odd-one-out on different configurations where RAISE picks out rule-breaking images interpretably via prediction errors on latent concepts. Figure 11 visualizes the experimental results of odd-one-out. RAISE will display larger prediction errors at odd concepts, which is important evidence when solving odd-one-out problems. It should be pointed out that forming such concept-level prediction errors requires the model to parse independent latent concepts and conduct concept-specific abstract reasoning correctly. RAISE can apply the atomic rules in the global knowledge set to tasks like out-one-out and has interpretability in generative abstract reasoning.

## D.5 STRATEGY OF ANSWER SELECTION

In this experiment, we evaluate RAISE with two strategies of answer selection: comparing candidates and predictions in pixel space (RAISE-pixel) and latent space (RAISE-latent). Table 7 reports higher accuracy when candidates and predictions are compared in latent space. Due to the noise in attributes, there can be multiple solutions to a generative RPM problem. Assume that the answer to an RPM is the image having two triangles, the answer images may significantly differ from each other in the pixel space by generating two triangles in various positions. However, they still point to the same concepts *Number*=2 and *Shape*=*Triangle* in the latent space. Therefore, selecting answers by comparing candidates and predictions in the latent space can be more accurate than comparing in the pixel space.

## E DISCUSSION ON BAYESIAN AND NEURAL CONCEPT LEARNING

**The learning objective.** A recent neural approach MLC (Lake & Baroni, 2023) uses meta-learning objectives to solve systematic generalization problems. Grant et al. (Grant et al., 2018) have reported a connection between meta-learning and hierarchical Bayesian models. The discussion section of MLC has also mentioned that the hierarchical Bayesian modeling can be explained from the view of meta-learning. In this perspective, the global atomic rules in RAISE act as global latent variables in hierarchical modeling. Although RAISE and MLC have different motivations for model design, there are potential connections and similarities between their learning objectives if we explain the reasoning process of RAISE from the perspective of hierarchical Bayesian modeling and meta-learning.

**Interpretability of latent variables.** Both Bayesian and neural approaches can define basic modules in the learning processes, e.g., Functions in Neural Interpreters (Rahaman et al., 2021) and atomic rules in RAISE. Bayesian approaches usually design interpretable latent variables in generative processes, e.g., RAISE uses categorical random variables to indicate the types of the selected rules explicitly. While Neural Interpreters route inputs to different Functions by calculating specific scores. DLVM provides a powerful learning framework to learn interpretable latent structures from data, e.g., RAISE defines latent concepts to capture image attributes. In this way, visual scenes are decomposed into a simple set of latent variables, which may reduce the complexity of abstract reasoning and enable systematic generalization on attribute-rule combinations.

**Solving multi-solution problems.** There can be multiple solutions for one generative reasoning problem due to the noise in data. DLVMs can handle multi-solution problems by stochastic sampling from the generative and inference processes. For example, RAISE can produce results different from the original sample but still follow the correct rules. Instead of making deterministic predictions, DLVMs attempt to provide probabilities of generating specific answers and capture randomness and uncertainty in abstract reasoning.

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

Table 8: **Answer generation at arbitrary positions on RAVEN**. We provide the average accuracy (%) and test errors (%) of ten trials on RAVEN for RAISE and baselines.

| | Center ($|T| = 1$) | | | | Center ($|T| = 2$) | | | |
|---|---|---|---|---|---|---|---|---|
| Model | $N_c = 1$ | $N_c = 3$ | $N_c = 7$ | $N_c = 15$ | $N_c = 1$ | $N_c = 3$ | $N_c = 7$ | $N_c = 15$ |
| LGPP | $55.8 \pm 2.5$ | $30.8 \pm 1.9$ | $17.1 \pm 2.0$ | $9.1 \pm 1.2$ | $53.8 \pm 1.4$ | $28.9 \pm 1.6$ | $15.4 \pm 1.2$ | $8.3 \pm 0.6$ |
| ANP | $61.4 \pm 0.7$ | $38.0 \pm 0.7$ | $23.5 \pm 0.9$ | $14.5 \pm 0.7$ | $58.3 \pm 0.5$ | $34.7 \pm 1.3$ | $20.5 \pm 1.0$ | $12.2 \pm 0.7$ |
| CLAP | $91.5 \pm 0.7$ | $80.1 \pm 1.6$ | $67.2 \pm 1.8$ | $53.8 \pm 1.7$ | $90.8 \pm 2.1$ | $80.3 \pm 3.4$ | $67.7 \pm 6.0$ | $55.3 \pm 6.2$ |
| Transformer | $99.6 \pm 0.2$ | $99.1 \pm 0.2$ | $98.5 \pm 0.3$ | $97.3 \pm 0.5$ | $97.2 \pm 2.3$ | $91.1 \pm 5.7$ | $88.0 \pm 4.0$ | $90.2 \pm 5.5$ |
| RAISE | $\mathbf{99.9 \pm 0.1}$ | $\mathbf{99.6 \pm 0.2}$ | $\mathbf{99.1 \pm 0.2}$ | $\mathbf{98.1 \pm 0.3}$ | $\mathbf{99.5 \pm 0.2}$ | $\mathbf{98.7 \pm 0.3}$ | $\mathbf{97.5 \pm 0.7}$ | $\mathbf{96.5 \pm 0.5}$ |

| | L-R ($|T| = 1$) | | | | L-R ($|T| = 2$) | | | |
|---|---|---|---|---|---|---|---|---|
| Model | $N_c = 1$ | $N_c = 3$ | $N_c = 7$ | $N_c = 15$ | $N_c = 1$ | $N_c = 3$ | $N_c = 7$ | $N_c = 15$ |
| LGPP | $56.8 \pm 2.3$ | $32.7 \pm 3.2$ | $18.7 \pm 2.0$ | $9.6 \pm 1.3$ | $57.4 \pm 2.1$ | $31.9 \pm 1.9$ | $18.4 \pm 2.0$ | $9.4 \pm 1.1$ |
| ANP | $59.0 \pm 0.6$ | $34.6 \pm 1.4$ | $20.6 \pm 0.9$ | $11.7 \pm 0.5$ | $60.5 \pm 1.2$ | $36.3 \pm 1.1$ | $21.7 \pm 0.9$ | $12.6 \pm 0.7$ |
| CLAP | $79.5 \pm 0.9$ | $60.7 \pm 1.2$ | $45.6 \pm 1.3$ | $32.4 \pm 0.9$ | $80.3 \pm 1.3$ | $62.6 \pm 2.6$ | $46.4 \pm 3.8$ | $33.9 \pm 2.6$ |
| Transformer | $99.4 \pm 0.2$ | $98.8 \pm 0.3$ | $98.1 \pm 0.4$ | $97.1 \pm 0.3$ | $95.8 \pm 1.6$ | $90.5 \pm 2.3$ | $87.2 \pm 2.8$ | $81.4 \pm 4.9$ |
| RAISE | $\mathbf{99.9 \pm 0.0}$ | $\mathbf{99.9 \pm 0.0}$ | $\mathbf{99.9 \pm 0.0}$ | $\mathbf{99.9 \pm 0.1}$ | $\mathbf{99.9 \pm 0.1}$ | $\mathbf{99.7 \pm 0.2}$ | $\mathbf{99.3 \pm 0.4}$ | $\mathbf{98.8 \pm 0.7}$ |

| | U-D ($|T| = 1$) | | | | U-D ($|T| = 2$) | | | |
|---|---|---|---|---|---|---|---|---|
| Model | $N_c = 1$ | $N_c = 3$ | $N_c = 7$ | $N_c = 15$ | $N_c = 1$ | $N_c = 3$ | $N_c = 7$ | $N_c = 15$ |
| LGPP | $57.5 \pm 2.3$ | $32.8 \pm 2.0$ | $19.7 \pm 4.1$ | $10.3 \pm 1.3$ | $57.6 \pm 1.5$ | $32.5 \pm 1.5$ | $18.0 \pm 1.2$ | $10.2 \pm 1.1$ |
| ANP | $58.3 \pm 1.1$ | $34.3 \pm 0.6$ | $19.4 \pm 0.8$ | $10.7 \pm 0.9$ | $59.6 \pm 0.6$ | $35.6 \pm 1.4$ | $20.8 \pm 0.5$ | $11.9 \pm 0.8$ |
| CLAP | $78.8 \pm 0.7$ | $59.1 \pm 1.2$ | $43.1 \pm 1.3$ | $30.2 \pm 1.1$ | $78.4 \pm 1.6$ | $59.9 \pm 2.8$ | $42.9 \pm 2.8$ | $31.5 \pm 2.8$ |
| Transformer | $98.9 \pm 0.2$ | $97.9 \pm 0.3$ | $96.5 \pm 0.4$ | $94.8 \pm 0.3$ | $92.3 \pm 1.7$ | $85.2 \pm 1.7$ | $75.6 \pm 3.1$ | $70.6 \pm 1.9$ |
| RAISE | $\mathbf{99.9 \pm 0.0}$ | $\mathbf{99.9 \pm 0.0}$ | $\mathbf{99.9 \pm 0.0}$ | $\mathbf{99.9 \pm 0.0}$ | $\mathbf{99.6 \pm 0.2}$ | $\mathbf{99.1 \pm 0.3}$ | $\mathbf{98.2 \pm 0.5}$ | $\mathbf{97.1 \pm 1.1}$ |

| | O-IC ($|T| = 1$) | | | | O-IC ($|T| = 2$) | | | |
|---|---|---|---|---|---|---|---|---|
| Model | $N_c = 1$ | $N_c = 3$ | $N_c = 7$ | $N_c = 15$ | $N_c = 1$ | $N_c = 3$ | $N_c = 7$ | $N_c = 15$ |
| LGPP | $50.5 \pm 1.3$ | $25.8 \pm 0.5$ | $13.2 \pm 0.7$ | $6.6 \pm 0.5$ | $49.8 \pm 1.3$ | $25.7 \pm 1.1$ | $12.8 \pm 0.5$ | $6.7 \pm 0.4$ |
| ANP | $62.0 \pm 1.2$ | $39.8 \pm 0.7$ | $26.5 \pm 0.6$ | $17.1 \pm 0.6$ | $61.6 \pm 1.1$ | $38.6 \pm 1.3$ | $24.3 \pm 1.2$ | $15.2 \pm 0.9$ |
| CLAP | $91.3 \pm 1.1$ | $81.1 \pm 1.8$ | $68.1 \pm 2.2$ | $54.1 \pm 2.2$ | $90.9 \pm 2.2$ | $81.4 \pm 2.4$ | $68.8 \pm 4.8$ | $57.5 \pm 6.5$ |
| Transformer | $97.6 \pm 0.4$ | $95.0 \pm 0.6$ | $90.1 \pm 0.5$ | $82.3 \pm 0.7$ | $96.7 \pm 1.7$ | $92.1 \pm 3.3$ | $90.2 \pm 3.8$ | $80.2 \pm 5.0$ |
| RAISE | $\mathbf{99.9 \pm 0.0}$ | $\mathbf{99.9 \pm 0.0}$ | $\mathbf{99.9 \pm 0.1}$ | $\mathbf{99.8 \pm 0.1}$ | $\mathbf{99.9 \pm 0.0}$ | $\mathbf{99.9 \pm 0.1}$ | $\mathbf{99.9 \pm 0.2}$ | $\mathbf{99.8 \pm 0.1}$ |

| | O-IG ($|T| = 1$) | | | | O-IG ($|T| = 2$) | | | |
|---|---|---|---|---|---|---|---|---|
| Model | $N_c = 1$ | $N_c = 3$ | $N_c = 7$ | $N_c = 15$ | $N_c = 1$ | $N_c = 3$ | $N_c = 7$ | $N_c = 15$ |
| LGPP | $49.9 \pm 0.6$ | $25.0 \pm 1.2$ | $12.0 \pm 0.6$ | $6.1 \pm 0.3$ | $50.0 \pm 0.9$ | $25.1 \pm 1.2$ | $12.1 \pm 0.7$ | $6.4 \pm 0.5$ |
| ANP | $66.1 \pm 1.1$ | $45.1 \pm 1.1$ | $30.1 \pm 2.0$ | $20.2 \pm 0.5$ | $66.5 \pm 1.0$ | $44.0 \pm 1.4$ | $28.5 \pm 0.8$ | $18.0 \pm 0.9$ |
| CLAP | $77.8 \pm 1.5$ | $58.4 \pm 1.8$ | $43.2 \pm 1.9$ | $30.5 \pm 1.0$ | $80.5 \pm 1.6$ | $63.1 \pm 3.4$ | $47.6 \pm 3.2$ | $35.2 \pm 2.4$ |
| Transformer | $97.9 \pm 0.4$ | $95.2 \pm 0.5$ | $90.6 \pm 0.9$ | $82.8 \pm 0.9$ | $93.2 \pm 1.7$ | $88.5 \pm 1.6$ | $80.4 \pm 3.7$ | $75.5 \pm 3.8$ |
| RAISE | $\mathbf{99.9 \pm 0.0}$ | $\mathbf{99.9 \pm 0.1}$ | $\mathbf{99.7 \pm 0.1}$ | $\mathbf{99.5 \pm 0.3}$ | $\mathbf{99.9 \pm 0.0}$ | $\mathbf{99.9 \pm 0.0}$ | $\mathbf{99.9 \pm 0.1}$ | $\mathbf{99.9 \pm 0.1}$ |

| | 2×2Grid ($|T| = 1$) | | | | 2×2Grid ($|T| = 2$) | | | |
|---|---|---|---|---|---|---|---|---|
| Model | $N_c = 1$ | $N_c = 3$ | $N_c = 7$ | $N_c = 15$ | $N_c = 1$ | $N_c = 3$ | $N_c = 7$ | $N_c = 15$ |
| LGPP | $51.3 \pm 1.0$ | $26.7 \pm 0.7$ | $13.6 \pm 0.8$ | $6.9 \pm 0.6$ | $52.6 \pm 1.4$ | $27.0 \pm 0.9$ | $13.8 \pm 0.7$ | $7.2 \pm 0.6$ |
| ANP | $54.8 \pm 0.9$ | $30.8 \pm 0.7$ | $18.2 \pm 0.5$ | $9.5 \pm 0.6$ | $55.5 \pm 1.0$ | $31.7 \pm 0.8$ | $18.4 \pm 0.8$ | $10.5 \pm 0.6$ |
| CLAP | $64.5 \pm 1.1$ | $39.9 \pm 1.5$ | $24.5 \pm 1.2$ | $15.0 \pm 0.9$ | $64.9 \pm 1.9$ | $41.8 \pm 1.5$ | $25.2 \pm 1.5$ | $16.9 \pm 1.4$ |
| Transformer | $64.3 \pm 1.2$ | $44.0 \pm 1.4$ | $30.3 \pm 1.5$ | $21.6 \pm 1.2$ | $63.1 \pm 1.0$ | $43.3 \pm 1.5$ | $28.8 \pm 1.4$ | $20.9 \pm 1.5$ |
| RAISE | $\mathbf{97.2 \pm 0.3}$ | $\mathbf{93.5 \pm 0.7}$ | $\mathbf{89.8 \pm 0.6}$ | $\mathbf{85.9 \pm 1.1}$ | $\mathbf{96.5 \pm 0.4}$ | $\mathbf{92.1 \pm 1.9}$ | $\mathbf{87.5 \pm 2.1}$ | $\mathbf{83.2 \pm 1.7}$ |

| | 3×3Grid ($|T| = 1$) | | | | 3×3Grid ($|T| = 2$) | | | |
|---|---|---|---|---|---|---|---|---|
| Model | $N_c = 1$ | $N_c = 3$ | $N_c = 7$ | $N_c = 15$ | $N_c = 1$ | $N_c = 3$ | $N_c = 7$ | $N_c = 15$ |
| LGPP | $53.2 \pm 1.3$ | $28.3 \pm 0.9$ | $14.8 \pm 0.4$ | $8.1 \pm 1.0$ | $52.8 \pm 1.2$ | $27.9 \pm 1.2$ | $14.8 \pm 0.9$ | $7.8 \pm 0.6$ |
| ANP | $53.9 \pm 1.0$ | $29.7 \pm 0.9$ | $16.7 \pm 0.2$ | $9.4 \pm 0.7$ | $55.0 \pm 1.2$ | $31.3 \pm 1.4$ | $17.9 \pm 0.7$ | $10.4 \pm 0.4$ |
| CLAP | $86.2 \pm 1.0$ | $71.2 \pm 1.3$ | $56.4 \pm 1.8$ | $43.9 \pm 1.2$ | $86.1 \pm 1.3$ | $72.3 \pm 2.8$ | $60.9 \pm 3.4$ | $47.1 \pm 4.0$ |
| Transformer | $59.4 \pm 0.8$ | $37.8 \pm 1.1$ | $24.3 \pm 0.8$ | $16.2 \pm 0.4$ | $59.5 \pm 0.8$ | $36.6 \pm 1.3$ | $23.6 \pm 0.8$ | $16.4 \pm 1.1$ |
| RAISE | $\mathbf{99.5 \pm 0.2}$ | $\mathbf{98.5 \pm 0.2}$ | $\mathbf{97.0 \pm 0.2}$ | $\mathbf{95.1 \pm 0.6}$ | $\mathbf{98.4 \pm 0.4}$ | $\mathbf{97.2 \pm 1.0}$ | $\mathbf{95.4 \pm 1.0}$ | $\mathbf{93.6 \pm 1.2}$ |

Table 9: **Answer generation at arbitrary positions on I-RAVEN**. We provide the average accuracy (%) and test errors (%) of ten trials on I-RAVEN for RAISE and baselines.

| | Center ($|T| = 1$) | | | | Center ($|T| = 2$) | | | |
|---|---|---|---|---|---|---|---|---|
| Model | $N_c = 1$ | $N_c = 3$ | $N_c = 7$ | $N_c = 15$ | $N_c = 1$ | $N_c = 3$ | $N_c = 7$ | $N_c = 15$ |
| LGPP | $55.0 \pm 2.9$ | $30.0 \pm 2.1$ | $16.8 \pm 1.6$ | $9.0 \pm 1.9$ | $54.3 \pm 1.3$ | $28.7 \pm 1.5$ | $15.7 \pm 1.4$ | $8.4 \pm 0.8$ |
| ANP | $77.1 \pm 1.2$ | $63.1 \pm 1.0$ | $53.0 \pm 1.1$ | $45.3 \pm 0.7$ | $64.5 \pm 0.8$ | $42.3 \pm 1.0$ | $28.0 \pm 0.8$ | $18.1 \pm 0.8$ |
| CLAP | $91.6 \pm 1.3$ | $79.6 \pm 1.3$ | $67.1 \pm 2.0$ | $53.4 \pm 1.9$ | $90.8 \pm 2.2$ | $80.8 \pm 4.6$ | $69.2 \pm 4.4$ | $51.7 \pm 4.2$ |
| Transformer | $99.8 \pm 0.1$ | $99.4 \pm 0.2$ | $98.9 \pm 0.3$ | $97.8 \pm 0.5$ | $95.2 \pm 2.2$ | $93.1 \pm 4.2$ | $87.6 \pm 7.9$ | $85.8 \pm 6.1$ |
| RAISE | $\mathbf{99.9 \pm 0.1}$ | $\mathbf{99.7 \pm 0.1}$ | $\mathbf{99.3 \pm 0.2}$ | $\mathbf{98.3 \pm 0.3}$ | $\mathbf{99.5 \pm 0.2}$ | $\mathbf{98.8 \pm 0.4}$ | $\mathbf{97.8 \pm 0.4}$ | $\mathbf{96.1 \pm 1.3}$ |

| | L-R ($|T| = 1$) | | | | L-R ($|T| = 2$) | | | |
|---|---|---|---|---|---|---|---|---|
| Model | $N_c = 1$ | $N_c = 3$ | $N_c = 7$ | $N_c = 15$ | $N_c = 1$ | $N_c = 3$ | $N_c = 7$ | $N_c = 15$ |
| LGPP | $57.1 \pm 2.9$ | $31.9 \pm 3.3$ | $19.4 \pm 2.3$ | $11.2 \pm 1.1$ | $56.7 \pm 1.9$ | $32.5 \pm 2.2$ | $18.7 \pm 1.7$ | $11.0 \pm 1.0$ |
| ANP | $71.4 \pm 1.0$ | $49.8 \pm 1.3$ | $35.4 \pm 1.0$ | $24.0 \pm 1.3$ | $68.7 \pm 1.3$ | $46.3 \pm 1.0$ | $30.8 \pm 0.8$ | $20.4 \pm 0.9$ |
| CLAP | $80.0 \pm 1.6$ | $61.5 \pm 1.3$ | $45.9 \pm 1.7$ | $33.3 \pm 1.3$ | $80.5 \pm 1.6$ | $63.2 \pm 2.6$ | $47.4 \pm 2.8$ | $35.1 \pm 2.5$ |
| Transformer | $99.7 \pm 0.1$ | $99.4 \pm 0.1$ | $99.0 \pm 0.2$ | $98.8 \pm 0.3$ | $96.4 \pm 1.3$ | $93.1 \pm 1.6$ | $89.3 \pm 2.5$ | $86.9 \pm 6.1$ |
| RAISE | $\mathbf{99.9 \pm 0.0}$ | $\mathbf{99.9 \pm 0.0}$ | $\mathbf{99.9 \pm 0.0}$ | $\mathbf{99.9 \pm 0.0}$ | $\mathbf{99.9 \pm 0.1}$ | $\mathbf{99.7 \pm 0.1}$ | $\mathbf{99.4 \pm 0.3}$ | $\mathbf{99.3 \pm 0.5}$ |

| | U-D ($|T| = 1$) | | | | U-D ($|T| = 2$) | | | |
|---|---|---|---|---|---|---|---|---|
| Model | $N_c = 1$ | $N_c = 3$ | $N_c = 7$ | $N_c = 15$ | $N_c = 1$ | $N_c = 3$ | $N_c = 7$ | $N_c = 15$ |
| LGPP | $58.5 \pm 2.5$ | $33.3 \pm 2.0$ | $19.2 \pm 3.8$ | $10.8 \pm 1.5$ | $56.1 \pm 2.6$ | $32.6 \pm 2.2$ | $19.6 \pm 2.0$ | $10.7 \pm 1.5$ |
| ANP | $69.5 \pm 1.4$ | $49.1 \pm 1.2$ | $33.8 \pm 0.9$ | $23.0 \pm 1.3$ | $66.7 \pm 1.1$ | $44.1 \pm 1.3$ | $29.2 \pm 1.1$ | $18.9 \pm 0.6$ |
| CLAP | $79.5 \pm 0.9$ | $60.4 \pm 1.5$ | $44.8 \pm 1.4$ | $32.3 \pm 1.4$ | $79.8 \pm 2.0$ | $59.9 \pm 2.0$ | $47.0 \pm 2.4$ | $32.0 \pm 2.4$ |
| Transformer | $99.5 \pm 0.1$ | $99.0 \pm 0.3$ | $98.5 \pm 0.4$ | $97.7 \pm 0.3$ | $94.8 \pm 1.2$ | $87.6 \pm 1.0$ | $82.0 \pm 3.1$ | $76.5 \pm 5.5$ |
| RAISE | $\mathbf{99.9 \pm 0.0}$ | $\mathbf{99.9 \pm 0.0}$ | $\mathbf{99.9 \pm 0.0}$ | $\mathbf{99.9 \pm 0.0}$ | $\mathbf{99.6 \pm 0.2}$ | $\mathbf{99.1 \pm 0.3}$ | $\mathbf{98.1 \pm 0.5}$ | $\mathbf{96.9 \pm 0.8}$ |

| | O-IC ($|T| = 1$) | | | | O-IC ($|T| = 2$) | | | |
|---|---|---|---|---|---|---|---|---|
| Model | $N_c = 1$ | $N_c = 3$ | $N_c = 7$ | $N_c = 15$ | $N_c = 1$ | $N_c = 3$ | $N_c = 7$ | $N_c = 15$ |
| LGPP | $51.4 \pm 1.0$ | $25.4 \pm 0.7$ | $12.9 \pm 0.9$ | $6.7 \pm 0.5$ | $50.5 \pm 1.3$ | $25.8 \pm 0.7$ | $13.1 \pm 0.7$ | $6.6 \pm 0.5$ |
| ANP | $81.5 \pm 0.7$ | $69.4 \pm 1.0$ | $59.5 \pm 0.9$ | $51.1 \pm 1.1$ | $71.6 \pm 1.2$ | $51.5 \pm 1.3$ | $37.9 \pm 1.6$ | $26.9 \pm 2.0$ |
| CLAP | $91.7 \pm 0.9$ | $81.4 \pm 1.3$ | $68.6 \pm 2.3$ | $57.8 \pm 4.3$ | $91.5 \pm 1.6$ | $82.3 \pm 2.7$ | $72.1 \pm 4.9$ | $57.1 \pm 3.7$ |
| Transformer | $99.1 \pm 0.2$ | $98.0 \pm 0.3$ | $95.9 \pm 0.4$ | $92.9 \pm 0.9$ | $97.9 \pm 1.5$ | $96.6 \pm 1.6$ | $94.2 \pm 2.5$ | $90.0 \pm 5.2$ |
| RAISE | $\mathbf{99.9 \pm 0.0}$ | $\mathbf{99.9 \pm 0.0}$ | $\mathbf{99.9 \pm 0.1}$ | $\mathbf{99.9 \pm 0.1}$ | $\mathbf{99.9 \pm 0.0}$ | $\mathbf{99.9 \pm 0.0}$ | $\mathbf{99.9 \pm 0.1}$ | $\mathbf{99.8 \pm 0.1}$ |

| | O-IG ($|T| = 1$) | | | | O-IG ($|T| = 2$) | | | |
|---|---|---|---|---|---|---|---|---|
| Model | $N_c = 1$ | $N_c = 3$ | $N_c = 7$ | $N_c = 15$ | $N_c = 1$ | $N_c = 3$ | $N_c = 7$ | $N_c = 15$ |
| LGPP | $50.0 \pm 1.3$ | $24.8 \pm 1.0$ | $12.6 \pm 0.6$ | $6.2 \pm 0.7$ | $49.7 \pm 0.7$ | $24.9 \pm 0.8$ | $12.4 \pm 0.6$ | $6.7 \pm 0.4$ |
| ANP | $82.6 \pm 0.7$ | $70.2 \pm 1.1$ | $60.3 \pm 1.2$ | $51.5 \pm 0.9$ | $75.9 \pm 0.8$ | $57.5 \pm 1.2$ | $42.1 \pm 2.4$ | $29.9 \pm 1.5$ |
| CLAP | $79.0 \pm 1.8$ | $60.2 \pm 1.1$ | $44.2 \pm 1.1$ | $32.2 \pm 1.9$ | $81.0 \pm 1.7$ | $64.2 \pm 1.3$ | $49.0 \pm 2.1$ | $36.4 \pm 2.6$ |
| Transformer | $99.0 \pm 0.3$ | $97.8 \pm 0.3$ | $95.6 \pm 0.4$ | $91.8 \pm 0.7$ | $96.6 \pm 0.9$ | $93.0 \pm 1.1$ | $87.9 \pm 2.5$ | $83.5 \pm 1.8$ |
| RAISE | $\mathbf{99.9 \pm 0.0}$ | $\mathbf{99.9 \pm 0.0}$ | $\mathbf{99.9 \pm 0.1}$ | $\mathbf{99.8 \pm 0.1}$ | $\mathbf{99.9 \pm 0.0}$ | $\mathbf{99.9 \pm 0.1}$ | $\mathbf{99.9 \pm 0.1}$ | $\mathbf{99.9 \pm 0.1}$ |

| | 2×2Grid ($|T| = 1$) | | | | 2×2Grid ($|T| = 2$) | | | |
|---|---|---|---|---|---|---|---|---|
| Model | $N_c = 1$ | $N_c = 3$ | $N_c = 7$ | $N_c = 15$ | $N_c = 1$ | $N_c = 3$ | $N_c = 7$ | $N_c = 15$ |
| LGPP | $51.3 \pm 0.8$ | $26.6 \pm 1.0$ | $13.9 \pm 0.5$ | $7.1 \pm 0.5$ | $51.9 \pm 1.0$ | $26.4 \pm 0.6$ | $13.7 \pm 0.4$ | $6.8 \pm 0.5$ |
| ANP | $54.9 \pm 1.0$ | $31.4 \pm 1.0$ | $18.0 \pm 1.0$ | $9.9 \pm 0.7$ | $55.6 \pm 0.7$ | $31.4 \pm 0.9$ | $18.4 \pm 1.0$ | $10.5 \pm 1.0$ |
| CLAP | $63.9 \pm 1.4$ | $40.2 \pm 1.5$ | $25.4 \pm 1.2$ | $14.8 \pm 1.0$ | $64.8 \pm 1.7$ | $42.5 \pm 2.0$ | $26.6 \pm 1.8$ | $16.1 \pm 2.1$ |
| Transformer | $65.7 \pm 1.4$ | $45.3 \pm 1.4$ | $32.1 \pm 1.1$ | $24.2 \pm 0.8$ | $64.2 \pm 0.3$ | $44.1 \pm 1.9$ | $31.5 \pm 1.4$ | $22.7 \pm 2.2$ |
| RAISE | $\mathbf{97.5 \pm 0.4}$ | $\mathbf{95.0 \pm 0.6}$ | $\mathbf{91.1 \pm 0.7}$ | $\mathbf{87.0 \pm 0.7}$ | $\mathbf{96.4 \pm 0.9}$ | $\mathbf{93.2 \pm 1.4}$ | $\mathbf{89.0 \pm 1.6}$ | $\mathbf{85.5 \pm 2.3}$ |

| | 3×3Grid ($|T| = 1$) | | | | 3×3Grid ($|T| = 2$) | | | |
|---|---|---|---|---|---|---|---|---|
| Model | $N_c = 1$ | $N_c = 3$ | $N_c = 7$ | $N_c = 15$ | $N_c = 1$ | $N_c = 3$ | $N_c = 7$ | $N_c = 15$ |
| LGPP | $52.4 \pm 1.3$ | $27.2 \pm 1.3$ | $14.9 \pm 1.4$ | $8.0 \pm 0.9$ | $52.4 \pm 1.0$ | $28.2 \pm 0.9$ | $15.0 \pm 0.8$ | $8.1 \pm 0.6$ |
| ANP | $54.2 \pm 1.2$ | $29.5 \pm 1.0$ | $16.6 \pm 0.8$ | $10.1 \pm 0.9$ | $54.5 \pm 1.1$ | $30.6 \pm 0.7$ | $17.4 \pm 1.1$ | $10.2 \pm 0.5$ |
| CLAP | $85.9 \pm 1.2$ | $71.7 \pm 1.3$ | $56.6 \pm 2.0$ | $42.6 \pm 1.5$ | $85.6 \pm 1.1$ | $70.4 \pm 3.0$ | $60.0 \pm 1.2$ | $46.4 \pm 3.0$ |
| Transformer | $59.7 \pm 1.3$ | $37.7 \pm 0.8$ | $25.0 \pm 0.7$ | $17.2 \pm 0.5$ | $59.7 \pm 1.0$ | $37.4 \pm 1.0$ | $24.5 \pm 1.1$ | $16.0 \pm 0.6$ |
| RAISE | $\mathbf{99.6 \pm 0.1}$ | $\mathbf{98.8 \pm 0.2}$ | $\mathbf{97.5 \pm 0.3}$ | $\mathbf{95.5 \pm 0.6}$ | $\mathbf{98.8 \pm 0.5}$ | $\mathbf{97.0 \pm 0.9}$ | $\mathbf{94.8 \pm 1.5}$ | $\mathbf{92.2 \pm 2.5}$ |

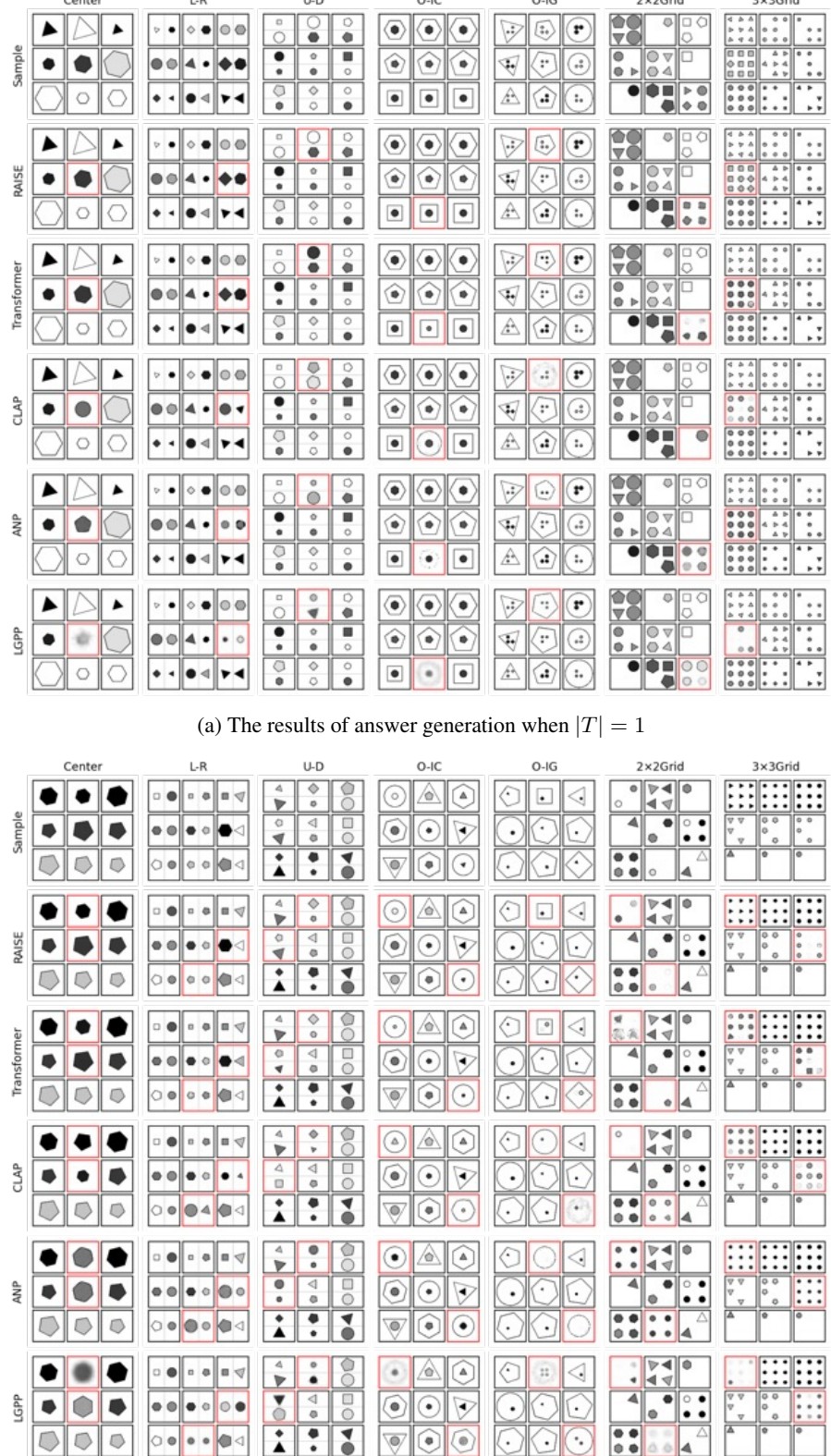

(a) The results of answer generation when $|T| = 1$

(b) The results of answer generation when $|T| = 2$

Figure 8: **Answer generation at arbitrary positions**. The predictions are given in red boxes to illustrate the ability of (a) arbitrary-position and (b) multiple-position answer generation.

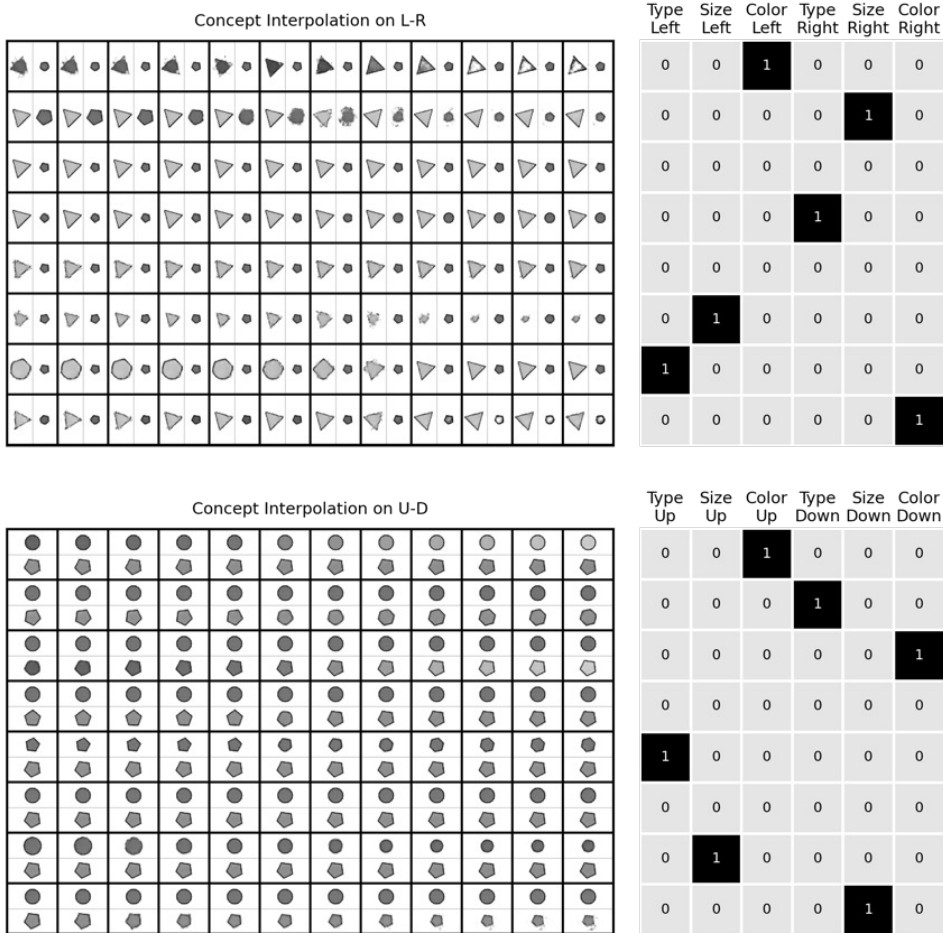

Figure 9: **Concept learning on RAVEN**. The table shows the interpolation results of latent concepts and the binary matrices indicating the correspondence between concepts and attributes on L-R and U-D.

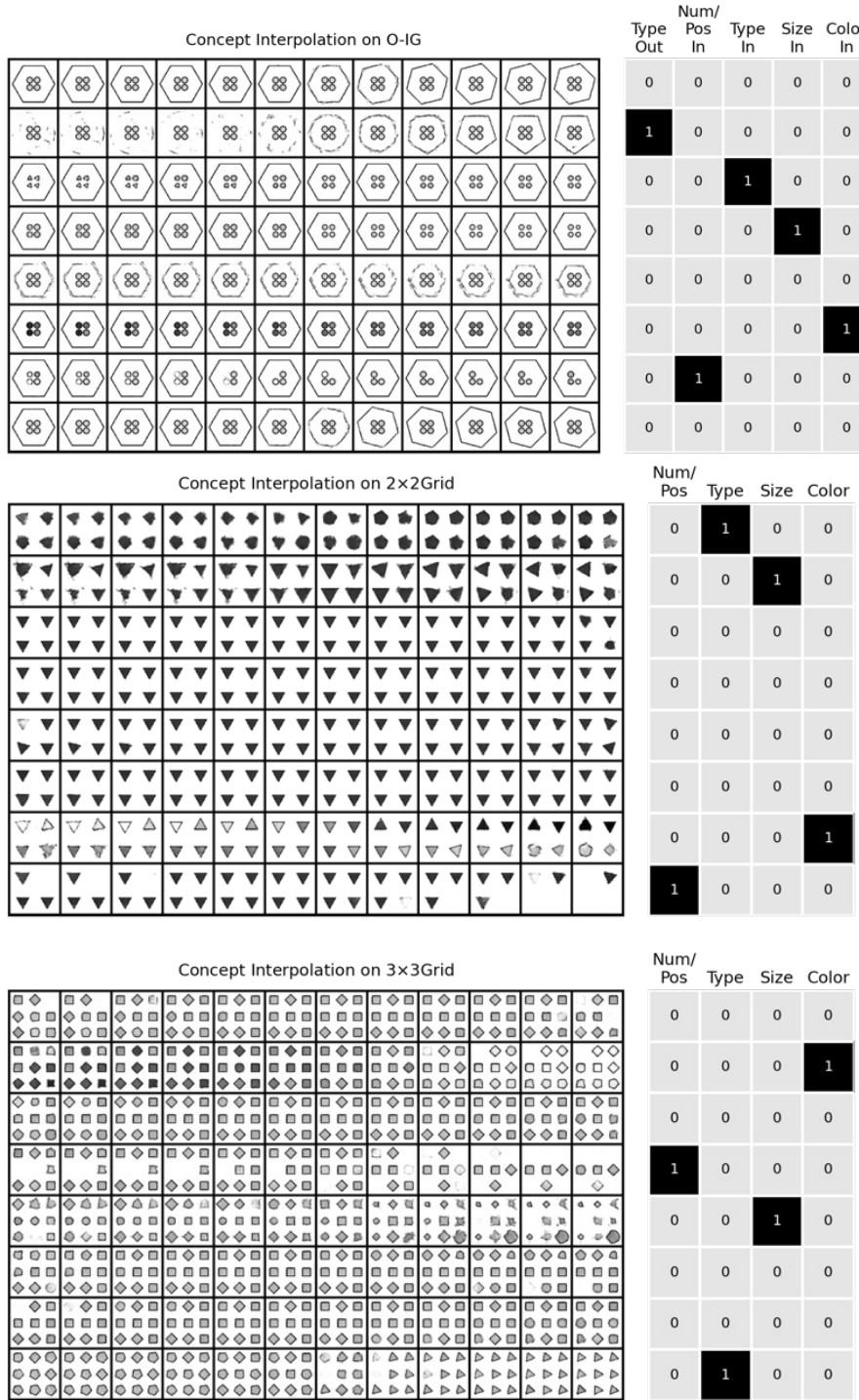

Figure 10: **Concept learning on RAVEN**. The table shows the interpolation results of latent concepts and the binary matrices indicating the correspondence between concepts and attributes on O-IG, 2×2Grid, and 3×3Grid.

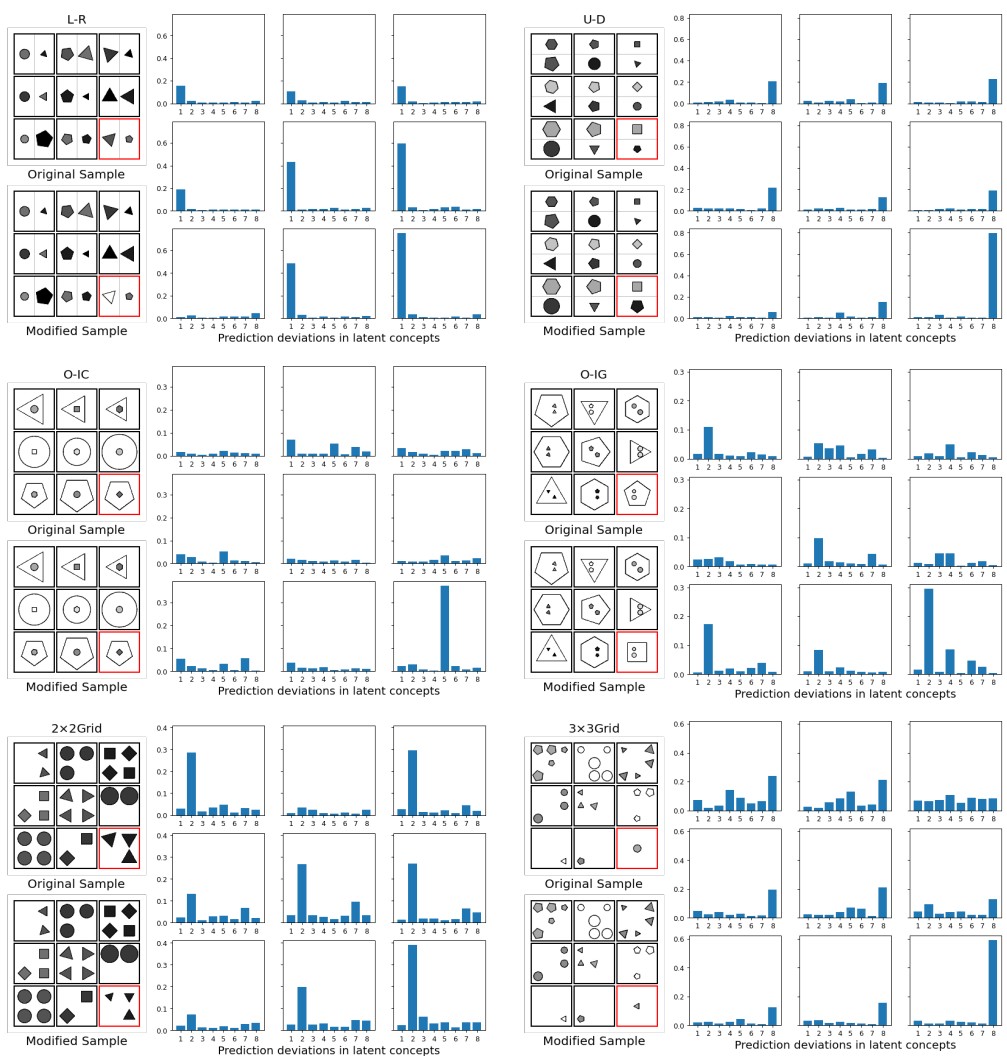

Figure 11: **Odd-one-out based on RPMs**. The plots display how to construct odd-one-out tests from different configurations of RPMs and how to find the odd image according to the prediction errors on latent concepts.

