# OpenReview forum: "Towards Generative Abstract Reasoning: Completing Raven’s Progressive Matrix via Rule Abstraction and Selection"
_ICLR.cc/2024/Conference — ICLR 2024 poster_

### Official Review · Reviewer_TA5q · 2023-11-01

**Soundness:** 3 good
**Presentation:** 3 good
**Contribution:** 3 good
**Rating:** 8
**Confidence:** 5

**Summary:**

The paper proposes RAISE, Rule AbstractIon and SElection model for generative abstract reasoning. In particular, the model is evaluated on Raven's Progressive Matrices. To solve a Raven problem, the model first encodes the context images and samples the latent rule for different latent concepts and generates the answer. On both RAVEN and I-RAVEN, the model shows improved performance while being generative, and also the model enables arbitrary panel generation and odd-one-out problem testing.

**Strengths:**

The best generative solution I'm aware of for solving the abstract reasoning problem, while problem structure has been taken into the design. To the best of my knowledge, previously similar performance is only attained by discriminative models and this is the first of the generative model to achieve similar performance. The authors have also tested the model on different setups, and the results are good enough.

The authors have shown sufficient experiments to show the meaningfulness of the latents: by varying the latents, they could get desired image results.

While the problem formulation follows the conventional approach, the problem decomposition makes sense and is intuitive.

**Weaknesses:**

The formulation for conditional generation is rather standard. While it is not exactly the same as conditional VAE, the derivation follows the same principles and the tweaks are only made due to the structured inference prior employed in modeling.

In general, I don't think the comparison is completely fair compared to other baselines, as some of the baselines only use the ground truth answers. While RAISE only uses the rule annotations, grounding of rules to corresponding hidden concepts is also implicitly encoded in the matrix. So supervisory signals could actually be backpropagated to the attribute / concept part. Besides, in evaluation, I do note that PrAE and ALANS are only trained on a specific split whereas RAISE are trained on more than one, and that is at least twice the data.

One thing I'm not particularly sure is how is the answer selected in RAISE. When you generate the answer, how do you pick the candidate from the given set? PrAE and ALANS actually only generate the hidden latents and compare in the latent space. Do you compare in the pixel space? Do you think comparing in the hidden space would help further improve performance of RAISE?

**Questions:**

Check above.

---

> ### Author Response · Authors · 2023-11-20
> **Response to Reviewer TA5q (Part I)**
>
> Thanks for the constructive suggestions. The detailed responses to the reviewer's comments are as follows.
>
> **Q1. In general, I don't think the comparison is completely fair compared to other baselines, as some of the baselines only use the ground truth answers.**
>
> The arbitrary-position solvers cannot leverage rule annotations since they do not explicitly model the category of rules. For a more fair comparison, we provide the selection **accuracy of unsupervised RAISE** in the following table.
>
> |          Models           |      Average      |      Center       |        L-R        |        U-D        |       O-IC        |       O-IG        |  2$\times$2Grid   | 3$\times$3Grid |
> | :-----------------------: | :---------------: | :---------------: | :---------------: | :---------------: | :---------------: | :---------------: | :---------------: | :------------: |
> |           LGPP            |     6.4/16.3      |     9.2/20.1      |     4.7/18.9      |     5.2/21.2      |     4.0/13.9      |     3.1/12.3      |     8.6/13.7      |   10.4/13.9    |
> |            ANP            |     7.3/27.6      |     9.8/47.4      |     4.1/20.3      |     3.5/20.7      |     5.4/38.2      |     7.6/36.1      |     10.0/15.0     |   10.5/15.6    |
> |           CLAP            |     17.5/32.8     |     30.4/42.9     |     13.4/35.1     |     12.2/32.1     |     16.4/37.5     |     9.5/26.0      |     16.0/20.1     | 24.3/**35.8**  |
> |        Transformer        |     40.1/64.0     | **98.4**/**99.2** | **67.0**/**91.1** |     60.9/86.6     |     14.5/69.9     |     13.5/57.1     |     14.7/25.2     |   11.6/18.6    |
> | **RAISE (no rule anno.)** | **54.5**/**67.7** |     30.2/56.6     |     47.9/80.8     | **87.0**/**94.9** | **96.9**/**99.2** | **56.9**/**83.9** | **30.4**/**30.5** | **32.0**/27.8  |
>
> We find that RAISE trained without rule annotations still achieves higher average accuracy than baselines.
>
>
> **Q2. While RAISE only uses the rule annotations, grounding of rules to corresponding hidden concepts is also implicitly encoded in the matrix. So supervisory signals could actually be backpropagated to the attribute / concept part.**
>
> The rule annotations may indicate the correspondence between attributes and rules, such as "Attribute #1 follows Rule #1". However, they **cannot** provide the meta information about attributes or rules (e.g., Attribute #1 is the number of objects, or Rule #1 is the progressive change). The annotations may inform RAISE how many independent attributes there are, but RAISE needs to automatically discover the meaning of attributes and encode them into latent concepts correctly.
>
>
>
> **Q3. In evaluation, I do note that PrAE and ALANS are only trained on a specific split whereas RAISE are trained on more than one, and that is at least twice the data.**
>
> We provide an additional experiment to train ALANS and PrAE in the same single-task setting as RAISE (i.e., the models are trained and tested on each configuration). The test results are given as follows.
>
> | Models |      Average      |    Center     |        L-R        |        U-D        |       O-IC        |       O-IG        |  2$\times$2Grid   |  3$\times$3Grid   |
> | :----: | :---------------: | :-----------: | :---------------: | :---------------: | :---------------: | :---------------: | :---------------: | :---------------: |
> |  PrAE  |     80.0/85.7     | 97.3/**99.9** |     96.2/97.9     |     96.7/97.7     |     95.8/98.4     |     68.6/76.5     | **82.0**/**84.5** |     23.2/45.1     |
> | ALANS  |     54.3/62.8     |   42.7/63.9   |     42.4/60.9     |     46.2/65.6     |     49.5/64.8     |     53.6/52.0     |     70.5/66.4     |     75.1/65.7     |
> | RAISE  | **90.0**/**92.1** | **99.2**/99.8 | **98.5**/**99.6** | **99.3**/**99.9** | **97.6**/**99.6** | **89.3**/**96.0** |     68.2/71.3     | **77.7**/**78.7** |
>
> In the single-task training setting, the accuracy of PrAE increases significantly. However, we found that using the official code, ALANS can hard converge to the accuracy provided in its original paper (the regular I.I.D. evaluation results). The experimental results show that, **in the same single-task training setting, RAISE achieves higher accuracies on most configurations and outperforms ALANS and PrAE on average accuracy**. We will replace the accuracy of ALANS and PrAE in Table 1 with the single-task results in the revised version.

---

> ### Author Response · Authors · 2023-11-20
> **Response to Reviewer TA5q (Part II)**
>
> **Q4. ... how do you pick the candidate from the given set? PrAE and ALANS actually only generate the hidden latents and compare in the latent space. Do you compare in the pixel space? Do you think comparing in the hidden space would help further improve performance of RAISE?**
>
> We select the candidate by comparing the latent concepts of the predicted target and candidate images. We conduct an experiment to illustrate the selection accuracy by comparing images and latent concepts. The experimental results are given in the following table.
>
> |     Models     |      Average      |      Center       |        L-R        |        U-D        |       O-IC        |       O-IG        |  2$\times$2Grid   |  3$\times$3Grid   |
> | :------------: | :---------------: | :---------------: | :---------------: | :---------------: | :---------------: | :---------------: | :---------------: | :---------------: |
> | RAISE (latent) | **90.0**/**92.1** | **99.2**/**99.8** | **98.5**/**99.6** | **99.3**/**99.9** | **97.6**/**99.6** | **89.3**/**96.0** | **68.2**/**71.3** | **77.7**/**78.7** |
> | RAISE (pixel)  |     72.9/77.8     |     95.2/96.8     |     90.6/95.8     |     96.6/98.5     |     80.4/90.6     |     69.1/81.1     |     40.1/42.6     |     38.1/39.5     |
>
>
> The results indicate that RAISE achieves higher accuracy by comparing latent concepts. Due to the existence of noise, there can be multiple solutions to a generative RPM problem. Assuming the answer is the image having two triangles, in this case, models can generate the two triangles in various positions. These two-triangle images **may significantly differ from each other in the pixel space**. While they **have the same concepts "Number=2" and "Shape=Triangle" in the latent space**. Therefore, RAISE selects answers by comparing candidates and prediction results in the latent space rather than the pixel space.

---

> > ### Comment · Reviewer_TA5q · 2023-11-23
> >
> > Thanks for the response. I'm satisfied by the new results and remain Accept for this work.

---

### Official Review · Reviewer_qkwH · 2023-11-01

**Soundness:** 3 good
**Presentation:** 3 good
**Contribution:** 2 fair
**Rating:** 6
**Confidence:** 4

**Summary:**

A new model -- RAISE has been proposed by the authors for RAVENs. The model contains several components, such as an image encoder, two variation autoencoder, and a global knowledge set. This model first extracted image features and then used these features to generate answers by a conditional generative process. The proposed model was evaluated through experiments on RAVEN and I-RAVEN datasets and showed better performance than other generative-based methods. Ablation studies showed the proposed RAISE can better handle the selection in arbitrary positions.

**Strengths:**

1. The paper is generally well-written and the main idea is easy to follow.

2. The authors proposed a new generative model -- RAISE that can explicitly encode rule-related information. It will be more useful to construct interpretable machine learning algorithms in this community.

3. The authors showed that their RAISE can perform better than previous generative methods, and is also on par with some of the existing selection-based methods.

4. The authors also analyze the proposed models with different experiments.

**Weaknesses:**

1. The authors fail to convince me, why we need to generate answers in arbitrary positions ? I am eager to see what is the advantage of generating an arbitrary position over generating the right-bottom answer, not only the final performance, but also the rationale or the motivation behind this design.

2. Whether the model shown in App is used in all configurations or not?

3. Too many hyper-parameters should be tuned.

4. Experiments are not enough. I suggest using PGM-Neutral, PGM-Interpolation and PGM-Extrapolaton to confirm the effectiveness of RAISE.

5. The authors also should clearly state why to use rule annotations, but not the answer images. Which indeed violates the RPM question.

Overall, it is a borderline paper

**Questions:**

See Weaknesses.

---

> ### Author Response · Authors · 2023-11-20
> **Response to Reviewer qkwH**
>
> Thanks for the constructive suggestions. The detailed responses to the reviewer's comments are as follows.
>
> **Q1. The authors fail to convince me, why we need to generate answers in arbitrary positions? I am eager to see what is the advantage of generating an arbitrary position over generating the right-bottom answer, not only the final performance, but also the rationale or the motivation behind this design.**
>
> We hope that RAISE can **understand the overall rule of RPM panels**, rather than learning to map given contexts to bottom-right images. The model's understanding of overall rules is illustrated through the ability to generate target images at arbitrary positions. Moreover, the arbitrary-position generation ability **can help RAISE solve abstract reasoning tasks besides RPM**. The experiment in Section 4.5 shows that RAISE can solve the odd-one-out problems by predicting the image at each position and comparing the prediction results with the realistic images to determine the rule-breaking image in the panel. If the model only predicts the bottom-right images, solving odd-one-out problems can be difficult as the rule-breaking images are not always located at the bottom-right.
>
>
>
> **Q2. Whether the model shown in App is used in all configurations or not?**
>
> Yes, all configurations of RAVEN/I-RAVEN use the same hyperparameters.
>
>
>
> **Q3. Too many hyper-parameters should be tuned.**
>
> In general, neural networks require inductive biases to learn attributes from images. Some generative solvers introduce artificially designed structures in representation learning, e.g., PrAE and ALANS design specific representations to encode attributes like object size, color, etc. While RAISE introduces inductive biases via hyperparameters on network structures and weights in the loss. By adjusting the hyperparameters, RAISE can adapt to different types of data and automatically encode attributes into latent concepts.
>
>
>
> **Q4. Experiments are not enough. I suggest using PGM-Neutral, PGM-Interpolation and PGM-Extrapolaton to confirm the effectiveness of RAISE.**
>
> We choose to evaluate models on RAVEN/I-RAVEN since their samples contain fewer noise attributes, and we can control the noise in RPMs to study the influence of noise in generative solvers by changing the settings of RAVEN/I-RAVEN, while PGM does not provide a similar way to control noise. We also refer to the configurations of PGM to generate two held-out configurations based on Center and O-IC and conduct experiments in Section 4.5 to evaluate the performance of models under OOD samples.
>
>
>
> **Q5. The authors also should clearly state why to use rule annotations, but not the answer images. Which indeed violates the RPM question.**
>
> RAISE uses images of correct answers to supervise the prediction results in training but does not require distractors in candidate sets. We think that there is no conflict between using candidate sets and rule annotations. The generative RPM solvers like PrAE and ALANS use both candidate sets and rule annotations in training. **Rule annotations can guide the generative solvers to generate target images with specific rules.** We also attempt to **reduce the use of rule annotations** by supporting semi-supervised training settings. RAISE does not use candidate sets for supervision, mainly because **it is trained via arbitrary-position generation, but the datasets do not provide candidate sets for non-bottom-right images**. We find that without using candidate sets, RAISE can generate better results than the compared solvers. During the testing phase, we evaluate the generative solvers in the conventional setting of RPM problems, which is to select answers from candidate sets based on the generated target images.

---

### Official Review · Reviewer_w266 · 2023-11-01

**Soundness:** 4 excellent
**Presentation:** 4 excellent
**Contribution:** 2 fair
**Rating:** 6
**Confidence:** 4

**Summary:**

This paper introduces a conditional generative model for Raven progressive matrices. It is trained from images, encoding each image into a set of C Gaussian latents, which can be decoded back to images, while also learning a set of K rules governing how a target image is composed (given its position in the grid and what the inferred progressive rule is). It is trained through an ELBO loss with some partial auxiliary rule supervision.

It shows clear advantages compared to existing baselines, both in terms of actual accuracy, but also in flexibility, as it can generate images in arbitrary cells of the RPM.

Overall, the generative model is rather straightforwardly designed, albeit feels rather specific to RPM problems, hence this is addressing a very specific problem and significance might be limited. I also have a few reservations about some baselines (in particular the Transformer one).

**Strengths:**

1. The paper is very clear, presents the problem well, and the math is clear and easy to follow. Figure 1 is very helpful to unpack the generative process, and overall I feel like it made all its choices clear. I could find nearly all details I needed about the implementational details (see questions below)
2. Experiments are comprehensive and well executed, with a good choice of baselines (I am not an expert in RPM however)
3. From results shown in Figure 2, I think it’s fair to say that it “solves” PGM quite effectively, but an expert might be able to comment better on the complexity of this problem for current SOTA.

**Weaknesses:**

1. This is a generative model designed specifically for RPM, and it is unclear how one would leverage this work or its findings in any other context.
2. It is equally a “straightforward” application of an ELBO-based generative process. It is well executed, but not surprising and I did not find particularly interesting pieces of technical/insightful choices throughout the paper.
3. It is unfortunate that some amount of rule annotation is still used. It oscillates between amounts (5% for non-grids, 20% for O_IG and 100% for 2x2 and 3x3 grids), but I was wondering what the performance would have been with 0% supervision, as this seems like the “optimal” solution target for RPMs.

**Questions:**

1. Do you have any suggestions for what one can learn from your model that can be generalized away from the RPM setting? Any specific insights / technical novelty compared to previous works?
2. Most decisions were extremely clear, but it was not that well explained how candidate answer selection was performed (Section 4.1 and 4.2)
   1. Do you generate a sample x_t and compare to the x_candidates in pixel space?
   2. Do you generate a z_t and compare g^enc(x_candidates)?
   3. Do you instead do a likelihood test?
3. Finding values for C and K required going into the Appendix (C=8, K=4), and their choice wasn’t discussed.
   1. How sensitive is the model to varying C or K?
   2. How were these chosen? How adapted to the number of attributes and real rule numbers do they have to be? I’m aware they are different, but how sensitive is it e.g. can you use K=100?
4. The Transformer baseline lacked details, even in Appendix C.3.
   1. Which encoder did you use? Was it a discrete representation? Transformers behave much better on VQ-like latents.
   2. I somehow expected it to do better, e.g. if attributes were provided instead of pixels, isn’t a Transformer a pretty strong baseline?
5. How needed was the rule supervision?
   1. Do you have numbers when you drop this to 0?
6. Nits/typos:
   1. It would have helped to explicitly write that r^c takes values between 1 and K.
   2. \mu_T^c -> \mu_t^c in (4)
   3. z_T^c -> z_t^c in (6)

---

> ### Author Response · Authors · 2023-11-20
> **Response to Reviewer w266 (Part I)**
>
> Thanks for the constructive suggestions. The detailed responses to the reviewer's comments are as follows.
>
> **Q1. How to generalize RAISE away from the RPM setting?**
>
> We provide three examples to illustrate the way to generalize RAISE away from the RPM setting.
>
> 1. RPM can be regarded as an **analogical reasoning task**. The first two rows are reference examples following specific rules, and models need to apply the rules to the third row to produce predictions (the third row is the problem panel where the first two images are queries, and the third image is the solution). RAISE can solve other analogical reasoning tasks by concatenating reference examples and problem panels into RPM-like matrices. The reasoning process of RAISE supports learning latent concepts and atomic rules on matrices of different shapes and generating solutions for given queries.
>
> 2. The **odd-one-out** experiment is another example. In this experiment, we leverage the arbitrary-position generation ability of RAISE to find the image that breaks the panel rules. We can set each image on the panel as the target image, generate the prediction of targets using RAISE, and find the one having the largest prediction error as the rule-breaking image.
>
> 3. Regarding image sequences as one-row matrices, we can use RAISE to **generate the missing frames in sequences**. For example, we can represent object appearances and positions in latent concepts, and different patterns of movement in atomic rules. RAISE can parse the movement pattern based on the given frames, and then apply it to the latent concept of position to generate the missing frames.
>
>
>
> **Q2.  Any specific insights / technical novelty compared to previous works?**
>
> The novelty of this paper lies in the definition of rules and how we realize atomic rules in the latent space via learnable networks. RAISE defines the **overall rules on the panel** instead of mapping the context images at eight positions to the bottom-right one or predicting the third image from the previous two images in a row. Therefore, RAISE **can predict answers at arbitrary and even multiple positions** conditioned on the given context and solve non-RPM reasoning tasks like odd-one-out.
>
>
>
> **Q3.  Details about candidate answer selection**
>
> RAISE computes distances between the generated target latent concepts and the latent concepts of candidates, and selects the candidate with the smallest distance.
>
>
>
> **Q4. Choice of $C$ and $K$**
>
> > How sensitive is the model to varying C or K? How were these chosen? How adapted to the number of attributes and real rule numbers do they have to be? I’m aware they are different, but how sensitive is it e.g. can you use K=100?
>
> To test the sensitivity of RAISE to $C$ and $K$, we train RAISE by varying $C$ ($C=4, 8, 16$) and $K$ ($K=4, 25$) on the original configuration. The results are given as follows.
>
> | Configs |  Average  |  Center   |    L-R    |    U-D    |   O-IC    |   O-IG    | 2$\times$2Grid | 3$\times$3Grid |
> | :-----: | :-------: | :-------: | :-------: | :-------: | :-------: | :-------: | :------------: | :------------: |
> |  $C=4$  | 59.5/78.6 | 98.8/99.7 | 42.2/79.1 | 34.4/62.0 | 51.3/85.7 | 37.0/65.1 |   72.6/80.4    |   80.2/78.0    |
> |  $C=8$  | 90.0/92.1 | 99.2/99.8 | 98.5/99.6 | 99.3/99.9 | 97.6/99.6 | 89.3/96.0 |   68.2/71.3    |   77.7/78.7    |
> | $C=16$  | 90.6/93.5 | 98.9/99.7 | 99.2/99.8 | 99.5/99.9 | 98.1/99.3 | 84.5/92.7 |   74.7/83.3    |   79.3/79.7    |
>
> | Configs |  Average  |  Center   |    L-R    |    U-D    |   O-IC    |   O-IG    | 2$\times$2Grid | 3$\times$3Grid |
> | :-----: | :-------: | :-------: | :-------: | :-------: | :-------: | :-------: | :------------: | :------------: |
> |  $K=4$  | 90.0/92.1 | 99.2/99.8 | 98.5/99.6 | 99.3/99.9 | 97.6/99.6 | 89.3/96.0 |   68.2/71.3    |   77.7/78.7    |
> | $K=25$  | 90.4/92.8 | 97.7/99.1 | 98.9/99.5 | 99.4/99.8 | 98.2/99.6 | 86.5/94.3 |   75.7/82.2    |   76.1/75.2    |
>
> RAISE is **insensitive** when increasing $C$ because it automatically generates some redundant latent concepts. When $C$ is too small to encode all changing attributes, the accuracy of RAISE will decline significantly (e.g., the accuracy of *L-R*, *U-D*, *O-IC*, and *O-IG* when $C=4$). To choose the proper $C$, we can first set a large number for $C$ and reduce it until the number of redundant latent concepts is reasonable. The results show that setting a large $K$ (here we set $K=25$ due to the limitation of devices) has **no significant influence** on the accuracy. But in general, we choose the hyperparameter $K$ by counting the unique labels in rule annotations.

---

> ### Author Response · Authors · 2023-11-20
> **Response to Reviewer w266 (Part II)**
>
> **Q5. Details about the Transformer baseline**
>
> > 1. Which encoder did you use? Was it a discrete representation? Transformers behave much better on VQ-like latents.
> >
> > 2. I somehow expected it to do better, e.g. if attributes were provided instead of pixels, isn’t a Transformer a pretty strong baseline?
>
> The Transformer baseline uses the same encoder and decoder structures as RAISE. We conduct **additional experiments** to evaluate the Transformer baseline with discrete representations. The results are given as follows.
>
> |        Models        |    Average    |      Center       |        L-R        |        U-D        |       O-IC        |       O-IG        | 2$\times$2Grid | 3$\times$3Grid |
> | :------------------: | :-----------: | :---------------: | :---------------: | :---------------: | :---------------: | :---------------: | :------------: | :------------: |
> |     Transformer      | 40.1/**64.0** | **98.4**/**99.2** |     67.0/91.1     |     60.9/86.6     | **14.5**/**69.9** | **13.5**/**57.1** | 14.7/**25.2**  | 11.6/**18.6**  |
> | Transformer-discrete | **44.2**/51.1 |     97.6/99.0     | **85.1**/**96.0** | **91.1**/**97.6** |     3.2/13.6      |     4.7/12.2      | **15.0**/22.2  | **12.7**/17.1  |
>
> We take the vector quantization method of VQ-VAE [1] to discretize the representations. We set the size of the discrete latent space as 512, the dimensionality of each latent embedding vector as 32, and the weight of commitment loss as 0.1. The outputs of the encoder are split into 8 vectors of the length 32 and replaced by looking up the nearest embeddings in the latent space. The experimental results show that **the discrete representations improve the performance on *L-R* and *U-D* but decrease the accuracy on *O-IC* and *O-IG* with in-out layouts**.
>
> We think the annotation of attribute values can be **too strong supervision** for model training. Though powerful RPM solvers like ALANS and PrAE leverage meta information about attributes to design visual representations, their training does not require any annotations of attribute values. We consider the Transformer as a baseline rather than a powerful solver because it is a black-box model and can hardly produce explicit representations of underlying rules on RPMs.
>
> [1] Van Den Oord, Aaron, and Oriol Vinyals. "Neural discrete representation learning." *Advances in neural information processing systems* 30 (2017).
>
>
>
> **Q6. How needed was the rule supervision? Do you have numbers when you drop this to 0?**
>
> Without the supervision of rule annotations, the accuracy of RAISE is
>
> |          Models           |  Average  |  Center   |    L-R    |    U-D    |   O-IC    |   O-IG    | 2$\times$2Grid | 3$\times$3Grid |
> | :-----------------------: | :-------: | :-------: | :-------: | :-------: | :-------: | :-------: | :------------: | :------------: |
> |        Transformer        | 40.1/64.0 | 98.4/99.2 | 67.0/91.1 | 60.9/86.6 | 14.5/69.9 | 13.5/57.1 |   14.7/25.2    |   11.6/18.6    |
> | **RAISE (no rule anno.)** | 54.5/67.7 | 30.2/56.6 | 47.9/80.8 | 87.0/94.9 | 96.9/99.2 | 56.9/83.9 |   30.4/30.5    |   32.0/27.8    |
> |           ALANS           | 74.4/78.5 | 69.1/72.3 | 72.2/79.2 | 73.3/79.6 | 76.3/85.9 | 74.9/79.9 |   80.2/79.5    |   75.0/72.9    |
> |           RAISE           | 90.0/92.1 | 99.2/99.8 | 98.5/99.6 | 99.3/99.9 | 97.6/99.6 | 89.3/96.0 |   68.2/71.3    |   77.7/78.7    |
>
> The experimental results show that, the average accuracy of unsupervised RAISE is between the unsupervised arbitrary-generation baselines and the generative RPM solvers trained with full rule annotations.
>
>
>
> **Q7. Typos**
>
> > 1. It would have helped to explicitly write that r^c takes values between 1 and K.
> > 2. \mu_T^c -> \mu_t^c in (4)
> > 3. z_T^c -> z_t^c in (6)
>
> Thanks for pointing out the typos. We will fix them in the revised version. We would like to clarify that, 2 and 3 are not typos. Here we use $T$ to index all target latent concepts.

---

### Official Review · Reviewer_5rkm · 2023-11-03

**Soundness:** 3 good
**Presentation:** 4 excellent
**Contribution:** 2 fair
**Rating:** 6
**Confidence:** 5

**Summary:**

The authors propose a novel method (RAISE) for solving Raven's Progressive Matrices (RPM) abstract visual reasoning problems. RAISE is a deep latent variable model, that learns the underlying rules in an RPM problem as latent variables and then uses the learned rules to conditionally (based on the context rows/columns present in one problem) generate the target image. The authors show that this leads to strong results for standard RPM reasoning tasks, generalizes better to more esoteric tasks (like answer selection at random locations). The visualization of latent concepts also shows that the latent variables are able to reasonably capture the atomic rules underlying the reasoning problem.

**Strengths:**

* The motivation of building a generative solver versus a selective one to avoid shortcuts in discriminative reasoning task is very valid and has been highlighted in prior literature [1, 2]. I commend the authors for trying to solve the harder problem in RPM reasoning: mapping the underlying data generative process.
* The RAISE model is very well thought our and describe in the paper. The modelling choices make sense for the few-shot style reasoning tasks in RPM problems. Also see weaknesses regarding the motivation and usability in a broader context.


**References**

1. Hu, S., Ma, Y., Liu, X., Wei, Y. and Bai, S., 2021, May. Stratified rule-aware network for abstract visual reasoning. In Proceedings of the AAAI Conference on Artificial Intelligence (Vol. 35, No. 2, pp. 1567-1574).
2. Geirhos, R., Jacobsen, J.H., Michaelis, C., Zemel, R., Brendel, W., Bethge, M. and Wichmann, F.A., 2020. Shortcut learning in deep neural networks. Nature Machine Intelligence, 2(11), pp.665-673.

**Weaknesses:**

* In Sec 1. Introduction: "It has been suggested that Bayesian inference with shared latent concepts can explain the emergence of in-context learning in LLMs, which is the foundation of few-shot reasoning in tasks like RPM tests." This premise has several logical leaps which the authors don't explain. For example, Chan et al [1] showed that the nature of training data for natural language enables few-shot reasoning from in-context learning. This does not hold for RPM reasoning tasks - the data distribution does not have the same long tailed and co-occurence properties as natural language. Similarly, Bayesian inference of shared latent concepts being able to explain in-content learning is just one lens of explainability for this empirical phenomenon, and there can be other lenses which are purely frequentist (e.g. the one showed by Chan et al). Overall, the motivation derived for RAISE from this statement is very loose and falls apart on deeper inspection.

* My chief concern witht this paper, and the overall idea behind structuring latent representations and re-combining them at inference is how scalable is it beyond toy problems like RPMs? The bottleneck with DLVMs like RAISE is two-fold: learnability (e.g. authors re-use encoder parameters between generative and inference process) and reliance on human knowledge to design the graphical model. RPMs have a very limited rule set, and the visual abstraction process is vastly simpler than human scenes. These reasons why existing concept learning methods that rely on similar modelling and inference have not been able to scale to harder visual reasoning problems are yet to be solved.

* In Sec 3.1 "Previous studies have emphasized the role of abstract object representations in the abstract reasoning of infants, which is similar to the idea of object-centric representation learners that decompose complex scenes into object representations. Both views reflect the compositionality of human cognition (Lake et al., 2011)." I don't know if the human analogy to concept learning should drive artificial concept learning agents, particularly because the underlying constraints (inductive biases?) on human and artificial systems are vastly different. A much more recent work from Lake et al [2] shows that neural network based reasoning combined with a meta learning objective can solve similar systematic generalization problems. I would also refer the authors to the discussion section of the paper for a much more nuanced discussion on Bayesian vs Neural approaches to learning systematicity. For a similar treatment of systematic reasoning by learning the underlying rule structure and routing at inference but from a purely neural treatment, I would refer the authors to [3] which also reports strong results on RPM style reasoning tasks.

**References**

1. Chan S, Santoro A, Lampinen A, Wang J, Singh A, Richemond P, McClelland J, Hill F. Data distributional properties drive emergent in-context learning in transformers. Advances in Neural Information Processing Systems. 2022 Dec 6;35:18878-91.
2. Lake, B.M. and Baroni, M., 2023. Human-like systematic generalization through a meta-learning neural network. Nature, pp.1-7.
3. Rahaman, N., Gondal, M.W., Joshi, S., Gehler, P., Bengio, Y., Locatello, F. and Schölkopf, B., 2021. Dynamic inference with neural interpreters. Advances in Neural Information Processing Systems, 34, pp.10985-10998.

**Questions:**

I don't have particular questions regarding the paper: I think it is a good paper, and the authors achieve what they set out to do in the introduction. My main concern is whether this is the right thing to do, especially for more "real-world" reasoning problems where the cardinality of the rule set is exponentially larger. I think the paper would benefit from the authors providing a more comprehensive discussion on Bayesian vs Neural concept learning since their current motivation on building a DLVM is not very convincing.

Minor question: In Sec 5. "Too much noise will make a problem have a large number of solutions (e.g., PGM (Barrett et al., 2018)), such data may not be proper for validating the generative reasoning ability of models" I am not exactly sure what the authors mean by "a problem have a large number of solutions" - does this refer to potentially many rule compositions satisfying the bottom-right answer selction in RPMs.

---

> ### Author Response · Authors · 2023-11-20
> **Response to Reviewer 5rkm**
>
> Thanks for the constructive suggestions. The detailed responses to the reviewer's comments are as follows.
>
> **Q1. Discussion on “Bayesian vs Neural concept learning”**
>
> We thank the reviewer for providing the insightful works on systematic generalization. We would like to discuss Bayesian and Neural concept learning in three folds.
>
> 1. **The learning objective.** As mentioned by the reviewer, MLC uses meta-learning objectives to solve systematic generalization problems. In the discussion of MLC, it is mentioned that the hierarchical Bayesian modeling can be explained from the perspective of meta-learning. Grant et al [1] also indicate the connection between meta-learning and hierarchical Bayesian models. From this perspective, the global knowledge of atomic rules in RAISE can be regarded as a kind of global latent variable in hierarchical modeling. Although RAISE and MLC have different thoughts on model design, there may be close connections between their learning objectives.
>
> 2. **Interpretability of latent variables.** Bayesian and Neural methods can define basic modules in the learning processes, e.g., Functions in Neural Interpreters and atomic rules in RAISE. In Bayesian methods, we usually take interpretable latent variables in the learning process, e.g., using categorical random variables to indicate the types of the selected rules. While Neural Interpreters route inputs to different Functions by calculating specific scores. We can define more latent variables with specific meanings in DLVMs, e.g., RAISE defines a set of latent concepts to capture attributes (object size, color, etc.). In this way, visual scenes are decomposed into a simple set of latent variables, which may reduce the complexity of abstract reasoning and enable systematic generalization on attribute-rule combinations.
>
> 3. **Solving multi-solution problems.**  In generative reasoning tasks, there can be multiple solutions for the given context of a problem. Bayesian methods can handle multi-solution problems by sampling latent variables from distributions (as shown in Figure 3, RAISE can produce results that are different from the original sample but still follow the correct rules). The experimental results of RAISE indicate that DLVMs can potentially solve generative RPM problems with multiple solutions in less noisy data.
>
> [1] Grant, Erin, et al. "Recasting Gradient-Based Meta-Learning as Hierarchical Bayes." *International Conference on Learning Representations*. 2018.
>
>
>
> **Q2. The bottleneck with DLVMs & Solving real-world reasoning problems**
>
> If we can design general inference and generative processes, DLVMs can potentially solve different tasks in various scenarios. RAISE achieves the arbitrary-position generation ability by decomposing high-dimensional images into low-dimensional latent concepts and learning atomic rules shared among latent concepts. This design allows RAISE to be potentially applied to different real-world scenarios.
>
> For example, when **predicting missing frames in surveillance videos**, latent concepts can represent object appearances (e.g., the appearance of a car) and spatial parameters (e.g., the position of the car), while atomic rules can encode different patterns of movement (e.g., turn left, stop, etc.). Through the given frames, RAISE can analyze the most likely movement pattern of the car, and then apply it to the latent concept of position to generate the missing frames. In this framework, latent concepts can also represent objects in scenes, and atomic rules can represent the actions learned from observation. We can apply the atomic rules to latent concepts (objects) parsed from new scenes and **predict the results after applying the action**.
>
>
>
> **Q3. What the authors mean by "a problem have a large number of solutions"**
>
> "A large number of solutions" here means there may be many solutions to the bottom-right answer. It is mainly caused by the noise in object attributes. Please refer to Figure 6 in the appendix.

---

### Author Response · Authors · 2023-11-21
**To all Reviewers**

Dear Reviewers,

We have uploaded the revised version of the paper. In the revised version, we
- fix the typos;
- add the description of the answer selection process in Section 4;
- replace the selection accuracy of ALANS and PrAE in Table 1 with the results in the single-task setting;
- add more details about hyperparameter choice and the Transformer baseline in Appendix C;
- add an experiment on different answer selection strategies in Appendix D.5.

---

### Meta-Review · Area_Chair_uaQy · 2023-12-04

**Metareview:**

This paper contributes a novel method for solving abstract visual reasoning problems and Ravens’ Progressive Matrices (RPM) in particular. The method itself is a deep latent variable model, whose latent structure reflects the inference process required for solving such tasks (eg. discovering rules, etc.). The choice of using a generative model, as opposed to a discriminative approach for this task stands out in particular. In terms of evaluation, clear advantages are showing compared to relevant baselines in terms of accuracy and generative capabilities. Other aspects of the model, such as the latent concepts are also studied.

The reviewers are in broad agreement that this is a paper worthwhile accepting. In particular the design of the generative model is highlighted, as well as the thoroughness of the evaluation. The paper is also well written and the problem that is being addressed is quite relevant. The main concern that was brought up is whether this architecture is too specific for RPMs and whether enough can be learned for abstract visual reasoning more broadly. I agree that this is a concern that affects significance, though there is also a clear precedent for publishing such works at conferences like ICLR and it is not unreasonable that others might learn from the insights that are presented here (independent of whether they directly carry over to an adjacent setting or problem). Overall the authors did a good job at addressing the remaining reviewer concerns/questions and I recommend accepting this paper.

**Justification For Why Not Higher Score:**

The significance and model design is somewhat limited to RPMs specifically and adjacent problems on analogical reasoning.

**Justification For Why Not Lower Score:**

All reviewers agree the paper is well executed, contributes (some) new insights, and the reported improvements are very signifiant with their particular niche.

---

### Decision · Program_Chairs · 2024-01-16

Accept (poster)